

# Toward continental hydrologic–hydrodynamic modeling in South America

Vinícius A. Siqueira[1], Rodrigo C. D. Paiva[1], Ayan S. Fleischmann[1], Fernando M. Fan[1], Anderson L. Ruhoff[1], Paulo R. M. Pontes[2], Adrien Paris[3,4,a], Stéphane Calmant[3], Walter Collischonn[1]

[1]Instituto de Pesquisas Hidráulicas, Universidade Federal do Rio Grande do Sul (UFRGS), Porto Alegre, 91501-970, Brazil
[2]Instituto Tecnológico Vale (ITV), Belém, 66055-090, Brazil
[3]LEGOS, Université de Toulouse, CNRS, CNES, IRD, UPS, Toulouse, France
[4]GET, Toulouse, France
[a]now at : Collecte Localisation Satellite (CLS), Ramonville-Saint-Agne, 31520, France

*Correspondence to*: Vinícius A. Siqueira (vinisiquera@gmail.com)

**Abstract.** Providing reliable estimates of streamflow and hydrological fluxes is a major challenge for water resources management over national and transnational basins in South America. Global hydrological models and land surface models are a possible solution to simulate the terrestrial water cycle at the continental scale, but issues on parameterization and limitations in representing lowland river systems put into question their utility for basin-scale analysis and to deliver daily discharges to meet local needs. In an attempt to overcome such limitations, we extended a regional, fully coupled hydrologic–hydrodynamic model (MGB) to the continental domain of South America and assessed its performance using daily river discharges, water levels from independent sources (in situ, satellite altimetry), estimates of terrestrial water storage (TWS) and evapotranspiration (ET) from remote sensing and other available global datasets. In addition, river discharges were compared with outputs from global models acquired through the eartH2Observe project (HTESSEL/CaMa-Flood, LISFLOOD and WaterGAP3), providing the first cross-scale assessment (regional/continental × global models) that makes use of spatially consistent daily discharge data. A satisfactory representation of discharges and water levels was obtained (NSE > 0.6 in 55 % of the cases) and MGB was able to capture patterns of seasonality and magnitude of TWS and ET especially over the largest basins of South America. Continental-scale modeling significantly improved discharge estimates when compared with global models, which resulted in a large number of gauges with negative (or close to 0) NSE values. Models were largely affected by positive bias mainly over East/Northeast Brazil and Argentina as well as over regions of Sao Francisco and Parnaiba basins, while major issues on flow timing were observed in regions affected by floodplain processes such as the Amazon, La Plata, Tocantins–Araguaia, Orinoco and Magdalena basins. We state that efforts in calibrating rainfall-runoff parameters within large basins are necessary to simulate daily river discharges appropriately in this continent, but implementing a hydrodynamic routing component is also important. We hope that our study provides further insights about hydrological simulation in South America, helping to reduce the gap between global and regional hydrological modeling communities.



**Keywords**: Model validation; Regional hydrological models; Global hydrological models; Cross-scale assessment; Discharge comparison; Manual calibration; GRACE.

## 1 Introduction

Reliable simulations of streamflow dynamics and related processes are vital to support water resources management regarding water security, natural hazards, navigation, agriculture and energy production. Therefore, improved predictions of the hydrological system can aid policymakers and stakeholders in making better decisions, also fostering actions to reduce risk and impacts on water resources under current and future conditions. In South America, recent important floods (e.g., Marengo et al., 2012; Hoyos et al., 2013; Ovando et al., 2016) and droughts (Melo et al., 2016; Erfanian et al., 2017),

together with uncertainties about the potential effects of climate change (Marengo et al., 2009) are encouraging new strategies for meeting social, economic and environmental water demands in large river basins all over the continent, some of them extending beyond political borders.

In this context, large-scale hydrological models arise as important tools for simulating the terrestrial phase of the water cycle. Despite limitations related to observed (in situ) data, especially in developing countries, advances in computational resources

and remote sensing technologies are pushing such models toward continental and global scales with increasing resolution (Wood et al., 2011; Bierkens, 2015; Bierkens et al., 2015; Sood and Smathkin, 2015). Currently, estimates of water fluxes at these scales are usually obtained using two modeling frameworks, namely global hydrological models (GHMs) and land surface models (LSMs) (Haddeland et al., 2011). While GHMs are more concerned with water resources assessment and lateral transfer of water, thus enabling quantification of human impacts and water scarcity/stress (e.g., Döll et al., 2009;

Wada et al., 2011), LSMs were primarily developed to provide lower boundary conditions for atmospheric circulation models, i.e., having a particular focus on vertical fluxes of heat and water (e.g., Balsamo et al., 2009). The latter are often coupled (i.e, in offline mode) to global river routing models designed for transporting water along drainage networks (e.g., Decharme et al., 2008; Zaitchik et al., 2010; Yamazaki et al., 2011; Getirana et al., 2012), which also enables the conversion from surface and groundwater runoff into river discharge and other surface water variables (e.g., flood extent, water level).

Although global-scale models can provide valuable spatiotemporal estimates of water fluxes and projections of those estimates (Sood and Smathkin, 2015), their ability to reproduce discharge observations at basin scale and to address practical water management issues is still limited (Archfield et al., 2015; Hattermann et al., 2018). Inaccuracies in runoff estimation from GHMs and LSMs may be first attributed to the uncertainty in global satellite precipitation products (Tian and Peters-Lidard, 2010; Sperna Weiland et al., 2015), but several studies have shown considerable differences between model outputs

even when using the same meteorological forcing, given the lack of knowledge about runoff generation processes and deficiencies in parameter estimation (e.g., Haddeland et al., 2011; Gudmundsson et al., 2012; Zhou et al., 2012; Beck et al., 2017a). In particular, calibration has been found to have the largest impact on storage fluxes, evapotranspiration and





discharge in comparison to variations in model structure and forcing data (Müller Schmied et al., 2014), which is a reason to call for efforts on this exercise as many of the GHMs and LSMs are not calibrated (Sood and Smathkin, 2015; Zhang et al., 2016; Beck et al., 2017a).

An alternative to overcome some limitations of GHMs and LSMs is to expand the spatial domain of hydrological models

that were initially developed for catchment to regional scales. Applying these models at national (e.g., Crooks et al., 2014) to continental domains (e.g., Abbaspour et al., 2015; Pechlivanidis and Arheimer, 2015; Donelly et al., 2016) translates into a better use of local expert knowledge and country-specific datasets that may be difficult to reach globally. At the same time, it is possible to focus on regionally relevant processes that are usually not included or not well resolved in global models. In South America, for example, several previous studies suggested that lateral water fluxes in large lowland rivers should be

resolved using hydrodynamic routing (e.g., Paiva et al., 2011, 2013; Paz et al., 2011, 2014; Yamazaki et al., 2011; Pontes et al., 2017; Zhao et al, 2017), while GHMs generally apply methods based on constant/variable velocity or a kinematic simplification of the St. Venant equations (see the overview by Kauffeldt et al., 2016 and Bierkens, 2015). Even if LSMs can be offline coupled to more physically based global river routing models (e.g., Yamazaki et al., 2011; Getirana et al., 2017b), calibration in the latter is likely to compensate for errors in runoff generation (Pappenberger et al., 2010; Getirana et al.,

2013; Hodges, 2013) and lack of relevant vertical hydrological processes linked to river–floodplain dynamics (e.g., Pedinotti et al., 2012; Paz et al., 2014; Fleischmann et al., 2018). In turn, fully coupled large-scale hydrologic–hydrodynamic models (e.g., Paiva et al., 2013) can handle the above interactions while using one single modeling framework, and are now feasible for using in continental domains because recent routing schemes (e.g., Bates et al., 2010) have proved to be computationally efficient for both regional (Getirana et al., 2017b; Pontes et al., 2017; Fleischmann et al., 2018) and global simulations

(Yamazaki et al., 2013).

Over the past decades, skill in streamflow prediction has been emphasized in catchment to regional-scale modeling (Archfield et al., 2015), but there is a growing opportunity to perform further spatial analyses rather than relying just on point measurements. Currently, a wide range of remote sensing products can be used to assess other variables than discharge, such as terrestrial water storage (e.g., Tapley et al., 2004; Watkins et al., 2015), evapotranspiration (e.g., Miralles et al.,

2011; Zhang et al., 2018), soil moisture (e.g. Kerr et al., 2012) or water surface elevation derived from satellite altimetry (e.g., Santos da Silva et al., 2010). Previous studies have shown the utility of the aforementioned datasets not only to validate hydrological/routing models (e.g., Alkama et al., 2010; Getirana et al., 2012; Paiva et al., 2013) but also as an interesting tool to constrain and estimate model parameters (Getirana, 2010; Werth and Guntner, 2010; Lopez et al., 2017). Therefore, remote sensing products can be helpful for continental-scale modeling in assessing regions where streamflow data are scarce,

as well as to outline areas in which future model improvements are potentially needed.

In parallel, the interest in building catchment/regional models up to continental domains, together with global models trying to be locally relevant through hyper-resolution (Wood et al., 2011), fosters the need to reduce the gap between these two modeling communities (Bierkens et al., 2015; Archfield et al., 2015). If the primary goal of a continental model is to provide river discharges to support regional water management demands at the basin scale, the results can also be used as a reference



to benchmark estimates from global models. In the last years, there has been an increasing number of studies assessing outputs of LSMs through multimodel intercomparison (e.g., Zaitchik et al., 2010; Gudmundsson et al., 2012; Zhou et al., 2012), sometimes in conjunction with GHMs (e.g., Haddeland et al., 2011; Beck et al., 2017a). Other recent studies focused on intercomparison of regional models in large basins around the world, usually relating overall performance to a single

gauge station and having a particular interest in monthly statistics (e.g., Huang et al., 2016; Eisner et al., 2017; Krysanova et al., 2017). Moreover, little attention has been given to intercomparison of global and regional-scale models, and the existing studies focused only on monthly to annual flows (e.g., Zhang et al., 2016) and projection of climate change impacts using a small number of gauge stations (e.g., Gosling et al., 2011; Hattermann et al., 2018). As streamflow is highly variable over space and at short time scales (i.e., daily), model performance should be assessed with spatially consistent data within large

basins and at sub-monthly intervals (e.g., Wu et al., 2014; Beck et al., 2017a; Zhao et al., 2017), but to our knowledge, no cross-scale (i.e., regional/continental × global models) intercomparison with a comprehensive evaluation of daily river discharges has been carried out over South America.

In this paper, we aim to start bridging this gap by (i) extending a regional-scale, fully coupled hydrologic–hydrodynamic model to the entirety of South America, assessing its ability to represent discharge and other hydrological variables across

the continent; (ii) investigating to what extent a continental-scale model can improve daily discharges when compared with global models and (iii) identifying the major issues that should be addressed for modeling discharges in this continent. The next sections provide a brief description of (i) the major river systems of South America, (ii) modeling approaches, (iii) datasets selected for validation, (iv) calibration procedures, (v) global models selected for discharge comparison and (vi) metrics used for assessment of results.

**2 Overview of the major South American river systems**

South America is one of the most freshwater abundant regions on Earth, contributing around 30 % of the global runoff to the oceans (Clark et al., 2015) despite having only 12 % of the total land area. Because of a combination of wide latitudinal extent (10º N–55º S), major orographic features and strong oceanic influences (Garcia and Mechoso, 2005; Vera et al., 2006; Garreaud et al., 2009), the continent is subject to a diverse climate that feeds six out of the 10 largest basins in the world in

terms of mean annual discharge, four of them only within the Amazon (Latrubesse et al., 2005). In particular, the Amazon is probably the most relevant hydrological system of the world, which draws great scientific attention due to its ecological importance and role in local to global climate (Werth and Avissar, 2002; Vera et al., 2006).

Figure 1 presents general information about South America, including hydroclimatic characteristics, major wetlands and hydrological regions. According to classifications of the Food Agriculture Organization (FAO) and Brazilian National Water

Agency (ANA), the continent is partitioned into 27 major hydrological divisions, with the largest ones (Amazon, La Plata and Orinoco basins) sharing much of their water between different countries. Because almost 80 % of the territory lies between the tropics, the continent is dominated by an equatorial (tropical) climate that drives high amounts of rainfall



especially near the equator, but extreme conditions also exist and range from very arid (such as the Atacama desert in the northern Chile) to polar areas over the Andes Cordillera and south of Patagonia (southern Argentina and Chile). As a result, water availability is subject to be highly variable with large discrepancies in mean annual flows, for example, 8.7 mm yr$^{-1}$ at Desaguadero River in the Colorado basin (Canalejas gauge station, 1.8 10$^5$ km²) compared with 2100 mm yr$^{-1}$ at Japura

River (Puerto Cordoba gauge station, 1.5 10$^5$ km²), the latter located in the Amazon basin.

(Figure 1)

Regarding river dynamics, flows in the Amazon are largely affected by floodplains over extensive flat terrains, causing

significant flood peak delay and attenuation (Richey et al., 1989; Alsdorf et al., 2007; Yamazaki et al., 2011; Paiva et al., 2013). Because flood waves have travel times in the order of a few months, sometimes out of phase because of the seasonal differences in precipitation (Richey et al., 1989), rivers are subject to strong backwater effects that extend for several hundred kilometers upstream of the river mouth or the confluence of its tributaries (Meade et al., 1991; Getirana and Paiva, 2013; Paiva et al., 2013), existing at both high- and low-water periods (Trigg et al., 2009). To the north, the Orinoco basin

shares some characteristics of the Amazon, such as unimodal flood pulse and low interannual variability of floodplain inundation, especially in the Llanos region (Hamilton et al., 2002). In addition, the high amplitude of mainstem water level — 14 to 16 m — produces backwater effects in its tributaries with strong hydrological gradients (Rosales et al., 2002).

At the heart of South America, the La Plata basin plays a major role in terms of agriculture, hydroelectricity and the economy in general, corresponding for almost 70 % of the combined Gross Domestic Product in countries such as Argentina,

Paraguay, Bolivia and Brazil (Vera et al., 2006; Barros et al., 2006). La Plata is mainly composed of the Parana and Paraguay Rivers and, to a lesser extent, the Uruguay River, which are completely different with respect to flow conditions. For example, the Upper Parana is largely regulated by reservoirs (about 50 % of mean annual flow, according to Su and Lettenmeier, 2009) and provides around 75 % of discharge up to the confluence with Paraguay river, despite the similar drainage area of both basins (~10$^6$ km²) (Barros et al., 2006). The Paraguay basin, on the other hand, is largely influenced by

one of the largest wetlands in the world — the Pantanal —, a complex anabranching river–floodplain system characterized by very gentle slopes that can be lesser than 1.5 cm.km$^{-1}$ (Tucci and Clarke, 1998; Berbery and Barros, 2002; Paz et al., 2011; Bravo et al., 2012). Much of the water stored in the floodplain at high water does not return to the channels during the drying phase and become available for evaporation and infiltration (Paz et al., 2014), so that flood waves are lagged by about 4–6 months (Tucci and Clarke, 1998; Hamilton et al., 2002), up to 60 % water volume can be lost (Gonçalves et al., 2011) and the shape of downstream hydrographs are strongly modified (Bravo et al., 2012). Moreover, huge areas on the right

overbank of the Paraguay River (Chaco Plain) have poorly defined drainage networks associated with alluvial megafans (Latrubesse, 2015), which makes this basin one of the most challenging regions for hydrological modeling in South America. The headwaters of La Plata also border important hydrological systems such as the Tocantins–Araguaia and Sao Francisco in a tropical wet–dry biome called Brazilian Cerrado (i.e., Brazilian savanna). The former is composed of Tocantins and



Araguaia Rivers, flowing parallel northwards until joining approximately 400 km upstream of the basin mouth near the Amazon Delta. While the Tocantins is marked by a cascade of large dams, the Araguaia River is much less altered and hosts the huge Bananal plain, which contributes up to 30 % of reduction in peak discharge due to floodplain inundation (Lininger and Latrubesse, 2016). Regarding Sao Francisco, two-thirds of the runoff is generated at the upper part of the basin (Allasia et al., 2006) and the mainstem crosses a semiarid region known as the "drought polygon," which affects several parts of Northeast Brazil including the Parnaiba basin. At the south of the continent, rivers flowing to the Atlantic Ocean correspond to less than 40 % of the area (i.e., > 60 % is related to endorheic basins) (Pasquini and Depetris, 2007) and their annual cycles usually show two maxima, one associated with the winter rainfalls and another to snowmelt during spring and early summer (Rivera et al. 2018).

# 3 Methods

## 3.1 MGB model

### 3.1.1 Model description

The MGB, *Modelo hidrológico de Grandes Bacias* (Large-Scale Hydrological Model) is a conceptual, semi-distributed, large-scale hydrological model first presented by Collischonn et al., (2007). The choice of MGB for this study was motivated by several past applications in South America (Allasia et al., 2006), which encompassed rapid response (e.g., Collischonn et al., 2005, Siqueira et al., 2016a) to markedly seasonal and often slow response basins (e.g., Bravo et al., 2012; Paiva et al., 2013; Fan et al., 2016; Pontes et al., 2017). In its most recent version, basins are divided into unit-catchments (Paiva et al., 2011; Pontes et al., 2017), each one containing a single river reach with associated floodplain and hydrological vertical water balance. Combinations of soil type and land use within each unit-catchment are categorized as Hydrological Response Units (HRUs). Water and energy budgets are computed independently for each HRU of each unit-catchment, and soil is depicted as a bucket model with a single layer. Canopy interception is represented in terms of leaf area index (LAI) and evapotranspiration (soil, plant transpiration and open water evaporation) is calculated using the Penman–Monteith equation. Surface runoff is produced using the variable contribution area concept following the Arno model (Todini, 1996), while groundwater and subsurface flows are computed, respectively, with linear and nonlinear functions according to water availability in the soil layer. Runoff from each one of the components (surface, subsurface and groundwater) is propagated to the stream network using linear reservoirs (i.e., hillslope routing). As MGB was primarily developed for tropical regions, snow processes are not represented in the current model version.

Flow routing in river channels can be computed using the Muskingum–Cunge method (Collischonn et al., 2007), one-dimensional full hydrodynamic (Paiva et al., 2013) or the local inertial method (Pontes et al., 2017). In this work, the MGB was applied with the inertial routing as described by Pontes et al., (2017), which uses the 1D version of the explicit local inertial approximation proposed by Bates et al., (2010). The routing structure of MGB is similar to that one described by



Yamazaki et al., (2011, 2013), i.e., the volume of water stored in a given unit-catchment is the only prognostic variable, while other variables such as flow depth and flooded area are diagnosed from the stored volume using floodplain profiles derived with sub-grid topography. The floodplain is treated as a simple storage model and the water level for a given time step is assumed to be constant along the entire unit-catchment. In addition, the model accounts for evaporation in floodplains

and infiltration from flooded areas to the unsaturated soil (Fleischmann et al., 2018), thus feedbacks between hydrological and hydrodynamic modules can be also represented (i.e., a two-way coupling approach). Further details on model water balance and flow routing equations are presented in Supplementary Material S1.

### 3.1.2 GIS processing

All geoprocessing steps were conducted with an adapted version of IPH-Hydro Tools package (Siqueira et al., 2016b), using

the 15 arcsec HydroSHEDS flow direction map (Lehner et al., 2008) as the main input. We chose the latter because it has received extensive corrections to address topological problems in flat areas and endorheic regions (Lehner et al. 2008) and has been successfully applied for river routing in other studies (e.g., Yamazaki et al., 2013; Zhao et al., 2017). An upstream area threshold of 1000 km² was adopted for the onset of drainage networks, while unit-catchments and river reaches were delineated using a fixed-length vector-based discretization of $\Delta x = 15$ km (see Supplementary Material S1.3 for details). This

length threshold was selected to ensure a balance between model stability and efficiency, resulting in an improved resolution when compared with configurations used by Yamazaki et al., (2013) and Getirana et al., (2017b) for river routing with the local inertial method at global and Amazon domains (grids with 0.25º resolution), respectively.

To estimate sub-grid floodplain topography, we first computed the Height Above Nearest Drainage (HAND) map (Rennó et al., 2008) using flow directions and drainage networks derived from HydroSHEDS together with the Bare-Earth SRTM v.1

Digital Elevation Model (DEM) (O'Loughlin et al., 2016), and further a floodplain profile was created at each unit-catchment relating HAND value, flooded area and water volume similar as done by Yamazaki et al., (2013). The Bare-Earth SRTM, resampled from 3 to 15 arcsec to match the HydroSHEDS resolution, was adopted to account for vegetation biases in floodplains since the C-band radar used by the original SRTM is not able to penetrate fully through the canopy (Carabajal and Hardling, 2005; Berry et al., 2007). Channel bed elevation at a given unit-catchment was estimated subtracting channel

bankfull depth from river bank height (i.e., the elevation at bankfull depth) (Yamazaki et al., 2011; Paiva et al., 2013), the latter also derived from Bare-Earth SRTM. However, one of the drawbacks of using an unconditioned DEM such as the Bare-Earth SRTM is the high level of noise affecting channel bank elevations, which need to be attenuated to avoid excessive inundation in low-relief areas. Instead of applying smoothing algorithms that modify the original DEM values (e.g., Paiva et al., 2011), a simple linear regression was fitted to DEM pixels located over drainage networks within each

unit-catchment (river reach). Channel bank heights were set as the smoothed elevation associated with the center pixel of each river reach, while the original DEM values remained unchanged, for example, when computing the HAND model and associated floodplain profiles (see Supplementary Material S1.5 and S1.6, for more details).





### 3.1.3 River hydraulic geometry

Because flow routing is very sensitive to river geometry (Yamazaki et al., 2011; Getirana et al., 2012; Paiva et al., 2013), channel parameters such as bankfull width and depth must be properly defined. However, detailed information about channel geometry is usually not available for large-scale basins and a very common approach is to adopt classic hydraulic geometry
relationships (HGs) (Leopold and Maddock, 1953) for specific sites according to drainage area or discharge (Decharme et al., 2008; Yamazaki et al., 2011; Getirana et al., 2012; Paiva et al., 2013; Luo et al., 2017; Pontes et al., 2017). Here, the global database of Andreadis et al., (2013) was used to set initial values of bankfull widths and depths, which were derived from two-year return period flows using the Global Runoff Database Center (GRDC) data and universal HGs obtained from several rivers around the world. In addition, regional HGs (Beighley and Gummaldi, 2011; Paiva et al., 2011, 2013, Pontes,
2016) and width estimates based on satellite imagery from Pontes (2016) were included to improve the global channel geometries of Andreadis et al., (2013) for Amazon and La Plata basins.

### 3.1.4 Model forcing

The Multi-Source Weighted Ensemble Precipitation — MSWEP v1.1 (Beck et al., 2017b) — was used as precipitation input to the rainfall-runoff module of the MGB model. This is a 3-hourly, global-scale dataset (0.25º resolution) that optimally
combines satellite, reanalysis and daily gauge data, and it has been evaluated with satisfactory results in a recent comparison of several precipitation datasets (Beck et al., 2017c). Regarding climate variables used to compute evapotranspiration, mean monthly data for the period 1961–1990 were retrieved from the Climate Research Unit — CRU Global Climate v.2 (New et al., 2002), which provides long-term climatologies of temperature, pressure, radiation and wind speed for all land areas at 10′ resolution.

### 3.1.5 Land use and soil data

Herein, we used the 400-m resolution HRU merged product (soil + land use) for South America (Fan et al., 2015), which is available at https://www.ufrgs.br/lsh. Basically, the soil map is a combination of the Brazilian database RADAMBrasil and the FAO Digitized Soil Map of the World and Derived Soil Properties, the latter included to account for areas lying outside Brazil. Land use classification was retrieved from the Global Land Cover map, which was generated using Envisat MERIS
fine-resolution (300 m) satellite imagery over the year of 2009. Regional land use maps of some Brazilian states were further included in the HRU merged product to improve level of detail.

### 3.2 Validation datasets

### 3.2.1 Discharge and water level data

Daily records of discharge were collected from several national hydrological services including: Agência Nacional de Águas
(ANA/Brazil: http://www.snirh.gov.br/hidroweb/), Operador Nacional do Serviço Elétrico (ONS/Brazil, Reservoir



naturalized flows: http://ons.org.br/), Instituto Nacional del Agua (INA/Argentina: http://bdhi.hidricosargentina.gob.ar/), Instituto de Hidrología, Meteorología y Estudios Ambientales (IDEAM/Colombia: http://www.ideam.gov.co/solicitud-de-informacion), Servicio Nacional de Meteorología y Hidrología (SENAMHI/Peru and Bolivia), Dirección General de Aguas (DGA/Chile: http://snia.dga.cl/BNAConsultas/) and other databases such as the Environmental Research Observatory for geodynamical, Hydrological and Biogeochemical control of erosion/alteration and material transport in the Amazon (ORE-HyBam: http://www.ore-hybam.org) and the GRDC (http://www.bafg.de/GRDC/, for Ecuador and NE South America). Based on expert knowledge, we masked out gauges heavily influenced by upstream reservoirs and included only gauges with more than 10 000 km² of drainage area. In a few cases, however, this threshold was lowered to include, at least, a small number of gauges due to the lack of available data. Short time series with less than five years of records were also excluded from analysis.

Satellite altimetry data were obtained from the THEIA/Hydroweb website (http://hydroweb.theia-land.fr/). Within this database, time series of water surface elevation (WSE) were extracted manually using the methodology presented in Santos da Silva et al., (2010) and are provided at virtual stations (VSs) where the satellite ground track forms a cross-over with the river network (~10–40 cm of water level accuracy). A total of 841 VS were found over the Amazon basin, 10 over the Orinoco basin and 29 over the La Plata basin. Data are derived from the observations of Envisat and Jason-2 for the period of 2002–2010 (35-day repeat orbit) and 2008–2010 (10-day repeat orbit), respectively. In situ stage data from ANA gauge stations were also obtained for the Brazilian territory and were filtered using the same criteria as discharge data.

### 3.2.2 Terrestrial water storage (TWS)

Launched in 2002, the Gravity Recovery and Climate Experiment (GRACE) measures temporal changes in the Earth's gravity field (Tapley et al., 2004). Several studies have shown the ability of GRACE to detect continental water storage variations at large spatial scales (e.g., Wahr et al., 1998; Ramillien et al., 2004; Tapley et al., 2004), which can provide insights into hydrological modeling about potential deficiencies in process description, parameters and input data (Schmidt et al., 2008). Here, we used the Release 05 JPL RL05M v2 mass concentration (mascon) estimates available at GRACE Tellus website (https://grace.jpl.nasa.gov). The JPL RL05M mascon solution solves for monthly gravity anomalies in terms of 3º × 3º equal-area spherical cap mascons, while using a Coastline Resolution Improvement (CRI) filter to discriminate between land and ocean mass portions of each mascon that spans coastlines (Wiese et al., 2016). Regarding the traditional spherical harmonic (SH) approach, which has been widely used in the last decade for global studies (Wahr et al., 1998; Landerer and Swenson, 2012), mascon solutions can also be applied to regional scales (Scanlon et al., 2016) and do not require the user to apply any postprocessing filters to the data, lowering the dependence on using scale factors to recover signal loss (Watkins et al., 2015). Uncertainties in GRACE mascon solutions (3º × 3º) over South America are around 10–15 mm of equivalent water thickness, and have been found to be similar or slightly lower in relation to SH solutions (Scanlon et al., 2016).



Despite of the native resolution (3º × 3º), mascon grids are provided with a spatial sampling of 0.5º × 0.5º. We kept the original resolution computing a simple average of 0.5º grid pixels located inside 3º × 3º mascon locations, as signals at sub-mascon resolution cannot be considered independent of each other. Time series of simulated TWS were first derived by summing water stored in all hydrological compartments, including rivers, floodplains, soil, groundwater and vegetation

canopy, at each time step. Similar to Paiva et al., (2013), the modeled TWS was then resampled as the weighted mean of TWS of all unit-catchments within each 3º × 3º equal-area mascon cell, using the former drainage area as weight. To ensure agreement with GRACE data, anomalies of simulated TWS were obtained by subtracting the long-term mean computed for the period between 2004 and 2009.

### 3.2.3 Evapotranspiration (ET)

Reference values of ET were extracted from the Climate Data Record (CDR) (Zhang et al., 2018), which is available at http://stream.princeton.edu:8080/opendap/MEaSUREs/WC_MULTISOURCES_WB_050/. Within this dataset, 10 gridded global ET products estimated from satellite (five), reanalysis (two) and LSMs (three) were optimally combined at 0.5º resolution using weighted averaging and a Bayesian merging technique. The weight of each product is related to the inverse of the ensemble spread and the deviation from the ensemble mean is assumed as a proxy of the uncertainty/error in

individual products. Together with other optimally merged variables provided by the CDR dataset (Precipitation, Runoff and TWS), estimates of ET were further adjusted with a Constrained Kalman Filter to ensure terrestrial water budget closure at each 0.5º grid cell (Zhang et al., 2018). For comparison purposes, the modeled ET was spatially aggregated into cells of 0.5º resolution using the unit-catchment drainage area as weight.

### 3.3 Model adjustment

Model calibration is commonly performed to improve agreement between observations and model results. However, the traditional gauge-by-gauge calibration used in regional hydrological modeling is not very common in continental to global domains (Archfield et al., 2015; Bierkens, 2015, Samaniego et al., 2017; Mizukami et al. 2017) because it can lead to spatial discontinuities of parameters (i.e., patchwork patterns) and overfitting to account for limitations in data and model structure. In other words, good results of discharge may not reflect a suitable depiction of the underlying hydrological processes, so

that modelers are more subject to "get the right answers for the wrong reasons" (Kirchner, 2006)

In an attempt to reduce the only dependency of river gauges during calibration, regions of parameter sets were derived by intersecting the global map of lithology/geology of Durr et al., (2005) with large South American basins/hydrological regions, the latter shown in Fig 1a. The Amazon and La Plata basins were further divided into their main tributaries prior to intersection due to their large spatial extent. As parameter sets do not correspond to a single gauge station, but rather to

30 regions defined by geological characteristics, multiple gauges were calibrated at the same time using the same parameter set. It is worth mentioning that calibration still remains a challenge for hydrological modeling with respect to large-scale



domains (Mizukami et al., 2017) and assessing the suitability of emerging parameter regionalization techniques (e.g., Samaniego et al., 2017) may be investigated in the future because it is beyond the scope of this paper.

The set of parameters used for model calibration is listed in Table 1, including their respective typical ranges of values. Other parameters used to compute energy balance and evapotranspiration (e.g., LAI, superficial resistance, albedo and canopy height) were defined *a priori* for each HRU vegetation type according to Collischonn (2001, and references within). Automatic calibration was not used herein to keep coherent values according to soil type and land cover, thus aiding to reduce model overparameterization. Therefore, sensitivity analysis of rainfall-runoff parameters was continuously performed as part of the manual calibration process. Regarding the hydrodynamic module, downstream boundary condition at oceans and lakes (endorheic basins) was set to a constant water level for simplification. Manning coefficient values were globally set to 0.03, with adjustments in specific rivers of the Amazon basin according to Paiva et al., (2013). Infiltration from floodplains to the soil column was considered and calibrated only for Pantanal region ($Kinf = 10$ mm day$^{-1}$) since previous studies showed that vertical hydrological processes largely influence model results in this area (e.g., Paz et al., 2014). It is worth mentioning that model sensitivity to river geometry and infiltration parameters were previously assessed by Paiva et al., (2013) and Fleischmann et al., (2018), respectively.

(Table 1)

As a result of the calibration procedure, several model parameter sets were manually adjusted and can be summarized into the following median values and percentile ranges (p5–p95): $Wm = 500$ (50–1500) mm; $b = 0.2$ (0.02–1.5); $Kbas = 0.2$ (0.01–3.0) mm day$^{-1}$, $Kint = 2$ (0.1–50) mm day$^{-1}$, $XL = 0.67$ (0.1–0.67), $Cs = 15$ (5–35), $Ci = 120$ (20–200) and $Cb = 1200$ (800–6000) h.

## 3.4 River discharge from GHMs and LSMs

Discharge outputs from state-of-the-art global models were acquired through the eartH2Observe Water Cycle Integrator (WCI, ftp://wci.earth2observe.eu). The WCI hosts multidecadal global water resources reanalysis datasets produced by 10 GHMs and LSMs, providing multi-scale (regional, continental and global) estimates of meteorological and hydrological water balance variables. We selected outputs from the 0.25° resolution Water Resources Re-analysis run 2 (WRR-2) baseline, which is an improved dataset over the WRR-1 (0.5°) produced by the initial project run (Schellekens et al., 2017). Models in the WRR-2 baseline are forced with MSWEP precipitation (1979–2014) and bias-corrected ERA-Interim data using the WFDEI correction methodology (see Martinez-de la Torre et al., 2018). Among the global models in WRR-2, river discharge at 0.25° resolution was available only for one LSM, the HTESSEL offline coupled to CaMa-Flood (Balsamo et al., 2009; Yamazaki et al., 2011), and two GHMs, namely, LISFLOOD (van der Knijf et al., 2010) and WaterGAP3 (Döll et al., 2009). The latter two have some degree of calibration and performed relatively well in terms of runoff in a recent model intercomparison (Beck et al., 2017a), while the former is uncalibrated but uses a state-of-the-art hydrodynamic routing




model. Within the eartH2Observe project, LISFLOOD and WaterGAP3 were run at, respectively, 0.1° and 0.08333°
resolutions, and discharges were then resampled to 0.25° for WRR-2 (Martinez-de la Torre et al., 2018). A brief overview of
the structure of these models is shown in Supplementary Material Table S2.1.

Because these models are grid-based, we followed a similar procedure to that in Zhao et al., (2017) to match grid cells to
5 corresponding river gauge stations. First, we applied an automatic routine to find the cell coordinates nearest to the gauge
locations. Cells were selected when the difference in the upstream area was within 5 %; otherwise, the surrounding cell with
minimum upstream area difference was selected. Gauges associated with cells whose drainage area differed more than 15 %
were excluded from the analysis. This procedure was performed separately for each global model to deal with differences
between their respective drainage networks. Moreover, due to the spatial resolution mismatch of LISFLOOD and
10 WaterGAP3, flow accumulation grids were recomputed using their respective flow direction maps (at 0.1° and 0.08333°) and
were resampled to the same resolution of discharge grids (0.25°). The corresponding cells were then extensively validated
with a thorough, GIS-assisted visual inspection, supported by long-term mean annual discharge grids (derived from each
global model) to minimize errors of gauge mislocation.

**3.5 Metrics for assessment of results**

MGB simulation was carried out between 01–Jan–1990 and 31–Dec–2009 using a daily time step and a warmup period of
two years to eliminate the influence of initial conditions. Model results were assessed in terms of discharge, water levels, ET
and TWS, while simulated river discharges were further compared with the output of global models. **Table 2** lists all
efficiency metrics used for assessment of model results. Statistics such as the Kling–Gupta Efficiency and Delay Index were
the same as used in Kling et al., (2012) and Paiva et al., (2013), respectively.

(Table 2)

**4 Results and discussion**

**4.1 Model validation**

**4.1.1 River discharges**

Simulated daily discharges were compared with in situ observations and results were presented in maps of performance
metrics ($r$, KGE and NSE) at each gauge station (Fig. 2). In addition, the runoff coefficient (RC = $Q_{mean}$/$P_{mean}$) was calculated
for each gauge station and was plotted against its respective KGE and drainage area (Fig. 3). There is a good agreement
between simulated and observed flows in several regions of South America, as NSE and KGE values are larger than 0.6 in
55 % and 70 % of the cases, respectively. Model performance is clearly higher in the southern and southeastern regions of
30 Brazil, including the central Amazon. On the other hand, performance decreases in regions marked by semiarid to arid
climates, such as in Northeast Brazil, west and southwest of the La Plata basin, most parts of Argentina and northern Chile.





For example, a poor correlation ($r < 0.2$) is observed in a semiarid region covered by the Colorado basin, where snow/glacier melt has a large contribution to total runoff and corresponds to the main source of water for human activities (Rivera et al., 2017). Other locations with lower performance (NSE < 0.2) usually refer to regions strongly influenced by orography (around Andes Cordillera), which are expected due to larger uncertainties of satellite-derived precipitation in these areas (Tian and Peters-Lidard, 2010; Paiva et al., 2013). River discharges at gauge stations with RC ranging between 0.3 and 0.6 are generally well represented in all spatial scales, while performance tends to be lower for RC < 0.3 and highly variable for rivers with lower drainage areas.

(Figure 2)

(Figure 3)

Figure 4 shows daily simulated discharges for some of the large South American rivers. The agreement between simulated and observed discharges is notable for both high and low flows in most of the cases, which indicates the model's ability to simulate regional to continental-scale rivers ($10^5$ km² to $4.7 \times 10^6$ km²) with different flow regimes. Results in the Amazon basin (e.g., Obidos, $NSE_{HD} = 0.89$) are comparable to other regional studies (e.g., Getirana et al., 2012; Paiva et al., 2013; Luo et al., 2017) while better performance is found over several of its tributaries (e.g., Purus, Madeira and Japura Rivers). Figure 4 also highlights the improvements of MGB using hydrodynamic (HD) over a non-hydrodynamic (noHD) routing method. In the Paraguay River, peak flows are dramatically reduced at the Amolar gauge station when using the HD routing (up to –75 %), and a similar behavior can be seen at Puerto Bermejo ($NSE_{noHD} = -5.8$ to $NSE_{HD} = 0.42$) located about 2400 km downstream near the confluence with the Parana River. Previous attempts of regional hydrological modeling in this basin that did not account for the floodplain inundation in the Pantanal (e.g., using the calibrated VIC model; see Su and Lettenmeier, 2009) reported negative NSE values for Puerto Bermejo, even at the monthly time scale. Differences in performance between noHD and HD are also quite remarkable (especially in terms of NSE) at gauge stations of Conceicao do Araguaia and Calamar in the lower Magdalena, where a pronounced attenuation effect is observed. On the other hand, in some rivers such as the Uruguay at Garruchos ($NSE_{noHD} = 0.85$; $NSE_{HD} = 0.82$), Parana at Itaipu ($NSE_{noHD} = 0.91$; $NSE_{HD} = 0.87$) and Tocantins at Descarreto ($NSE_{noHD} = 0.72$; $NSE_{HD} = 0.70$) the routing method has a minor impact. In the case of Orinoco at Ciudad Bolivar, both hydrographs look similar, but NSE suggests that results are improved when HD routing is used ($NSE_{noHD} = 0.83$; $NSE_{HD} = 0.9$).

(Figure 4)



### 4.1.2 Water levels

Performance metrics regarding water levels are presented in Fig. 5. For a suitable comparison, observed data and modeled WSE were first converted into anomalies (i.e., by subtracting their respective long-term mean: $h_{new} = h - \bar{h}$) to keep values with the same reference. In addition, Fig. 6 shows time series of simulated water level anomalies (hereafter referred to as water levels) for some of the large rivers of South America, which were plotted against in situ water levels and satellite altimetry. In general, the results obtained for the assessed gauge and VS stations are considered satisfactory in terms of correlation ($r \geq 0.8$ in 80 % of cases) and Nash–Sutcliffe (NSE $\geq 0.6$ in 60 % of cases), with a reasonable performance for amplitudes ($–30 \% < \sigma BIAS < 30 \%$ in 50 % of the cases). Similar to prior regional studies (e.g., Getirana et al., 2012, 2017b; Paiva et al., 2013), water levels are well represented in the central Amazon, where a good performance is observed along the Solimoes up to the river mouth in the Atlantic Ocean. Amplitudes are overestimated ($\sigma BIAS > 30 \%$) in southeast tributaries such as Madeira, Xingu and Tapajos, as well as in headwaters located in the northwest part of the basin. Outside the Amazon, there are acceptable results in the Orinoco (e.g., lower Meta), Uruguay and Tocantins–Araguaia basins, where the model generally performs well in all assessed metrics. Large overestimations in the standard deviation ($\sigma BIAS > +50 \%$) are systematically found over the Sao Francisco main stem, which are reflected by very low values of NSE ($< 0.2$). On the Paraguay River, a reasonable agreement between observed and simulated water levels is observed at Amolar, but performance significantly reduces in both correlation and $\sigma BIAS$ for downstream regions (e.g., at Porto Murtinho). In the latter case, model results are clearly advanced in time and are not capturing rapid variations of water level originating from lateral contributions of tributaries.

(Figure 5)

(Figure 6)

Performance in water levels is directly related to the agreement between simulated and observed discharges. On the other hand, even if discharges are well represented, there are uncertainties related to Manning values and also to river widths and depths derived from HGs, which do not reflect singularities of cross sections such as narrowing or widening of rivers at both gauge and VS locations. Previous studies have demonstrated the large influence of channel geometry and roughness on both amplitude and timing of water levels, especially over the Amazon basin (e.g., Yamazaki et al., 2011; Paiva et al., 2013; Paris et al., 2016; Luo et al., 2017). Moreover, river bed profiles are subject to DEM errors that can hardly be reduced through simple profile-smoothing procedures. For example, datasets used to remove the vegetation bias in Bare-Earth SRTM (IceSAT, vegetation height maps, uncorrected SRTM) have different spatial resolutions that lead to artifacts around the edges of vegetation patches (O'Loughlin et al., 2016), producing additional noise on river bed elevations. In addition to vegetation, other SRTM error sources such as stripe noise (Rodriguez et al., 2006) significantly affect large flat areas on the



La Plata basin (Yamazaki et al., 2017), which can ultimately impact model results. Model resolution and the ability to route discharge in downstream multi-directions (e.g., rivers with bifurcations and anabranching networks) can affect simulated water levels and flooded areas (e.g., Mateo et al., 2017), which has been taken into account in recent studies with MGB (e.g., Pontes et al., 2017; Fleischmann et al., 2018) but not in this continental model application. Other model assumptions like an

approximation of rectangular channels and ineffective flow over floodplains may also affect the results.

### 4.1.3 Evapotranspiration (ET)

Figure 7 shows the magnitude and seasonality of ET averaged for major basins in South America, as well as the magnitude of errors (RMSE) comparing modeled values to the optimal estimate of the CDR dataset. The ratio of the RMSE to the CDR uncertainty ($RMSE_{unc}$) was calculated to outline regions where simulated ET tends to deviate from the optimal CDR value,

i.e., values above unity indicate that the model error is larger than the mean deviation of all datasets (used in CDR) from their ensemble mean.

Results show that MGB can capture patterns of ET over the South America region. Simulated ET values are within the CDR uncertainty range in most of the continent ($RMSE_{unc}$ < 1), with errors varying between 10 and 30 mm month$^{-1}$. A good agreement in terms of magnitude and seasonality of ET is observed for the Amazon, Tocantins–Araguaia and mainly for La

Plata basin, but the model also performs reasonably well in Sao Francisco. Conversely, larger deviations ($RMSE_{unc}$ > 1) are found at low latitudes (20º S–10º N) where RMSE values reach up to 50 mm month$^{-1}$. ET is underestimated during the dry season in basins such as Orinoco (DJF), Amazon and Tocantins–Araguaia (JJA), while it is largely overestimated from the onset to the end of the dry (wet) season in Sao Francisco and Parnaiba (Orinoco). The latter two are clearly affected by a temporal lag in ET seasonality, where simulated values are delayed by approximately one month with respect to CDR

estimations. At mid latitudes (> 20º S), large RMSE values (above 50 mm month$^{-1}$ and $RMSE_{unc}$ > 2) are observed only in a narrow N–S range over the southern Andes.

Regarding issues on timing and magnitude of simulated ET, meteorological forcing probably has an influence on model performance because long-term mean climate data are used for ET computations. Another possible reason is the lack of spatial variability of moisture in the MGB soil column. Guswa et al., (2002) compared a simple bucket (one layer) to a

physically based Richards model, showing that large discrepancies can occur with respect to the relationship between ET and average root-zone saturation, as well as in timing and intensity of transpiration, especially for water-limited conditions. Moreover, Wang et al., (2006) found that time scales of evapotranspiration can differ significantly between a single-layer and multi-layer soil scheme due to nonlinear interactions that occur in the latter. Indeed, ET is expected to respond quickly at the beginning of the rainy season due to an increase of water availability at the soil surface layer, which cannot be well

represented with a single-layer, bucket-type model like MGB. In contrast to some of the datasets used in CDR (e.g., reanalysis and LSMs), the MGB does not account for snow processes, which may explain the large RMSE values over the southern Andean region.



(Figure 7)

Although it is beyond the scope of this study for a full assessment of ET estimations derived from hydrological models and
other sources, it is important to note that errors presented here correspond to the difference between both estimates (MGB
and CDR). ET is one of the most uncertain water balance variables due to its high spatial and temporal variability, thus it is
difficult to validate given the lack of ground observations (Miralles et al., 2011; Zhang et al., 2018). Even accounting for ~70
% of the weight in CDR dataset in comparison to LSM and reanalysis (Zhang et al., 2018), remote sensing products of ET
are based on Penman–Monteith or Priestley–Taylor equations that depend on vegetation indices and meteorological forcing
derived from satellite/reanalysis data, which are associated to many uncertainties (Miralles et al., 2011; Vinukollu et al.,
2011). Christoffersen et al., (2014) and Maeda et al., (2017) showed that most remote sensing and land surface models are
unable consistently to reproduce ET seasonal cycles in tropical areas (across the Amazon basin) when compared with eddy
covariance measurements and ET estimates from water balance. In the Amazon, for example, ET seasonality is regulated by
radiation, rainfall and how vegetation assimilates water and energy (Restrepo-Coupe et al., 2013; Maeda et al., 2017). Other
limitations are also associated with the vegetation cover fraction and how ET is partitioned between transpiration, soil and
canopy evaporation (Miralles et al., 2011).

### 4.1.4 Terrestrial water storage (TWS)

Figure 8 shows the performance of simulated TWS anomalies in comparison to observations from GRACE mascon
solutions. To evaluate the ability of MGB to reproduce monthly variations of TWS, both simulated and observed time series
of TWS were averaged to the scale of large basins in South America and are presented in Fig. 9.
In general, the results show that MGB has the ability to represent TWS anomalies over the continent. There is a good
temporal correlation in most part of tropical South America ($r > 0.75$), as well as in temperate regions with dry summer
between latitudes of 30º S and 40º S. Amplitudes of TWS are reasonably well simulated ($-20$ % $<$ σBIAS $< 20$ %) mainly in
central Brazil, parts of the northeast, south of La Plata and areas of South Chile. On the other hand, performance typically
decreases in semiarid to arid climates, as can be seen in regions such as North Chile, Colorado basin, west of La Plata and
South Argentina. High negative bias with large RMSE ($> 150$ mm) is observed in the northeast Amazon and west of the
Orinoco, whereas large overestimations are found mainly over coastlines at low latitudes in the southern hemisphere (0–20º
S). Moreover, in some regions characterized by polar climate, at the extreme south or in areas over the Andes, modeled TWS
anomalies are markedly underestimated (σBIAS $< -80$ %).
Modeled TWS is in good agreement with GRACE observations over the Amazon, Tocantins–Araguaia, Sao Francisco and
Parnaiba basins ($r > 0.9$, |σBIAS| $< 15$ % and RMSE $< 45$ mm), capturing both the interannual variability and amplitude of
TWS anomalies for the analyzed period. MGB is also successful in representing annual changes of TWS in the La Plata
basin, but with an overestimation (σBIAS $= 22$ %) probably caused by high positive σBIAS in the Paraguay and Chaco





regions. Errors in the La Plata basin (RMSE = 24 mm) are in the same order as those in the Amazon (RMSE = 26 mm). In addition, larger amplitude differences clearly occur in the Orinoco (σBIAS = –32 %), but with a pronounced RMSE (> 60 mm) that mainly originates from the eastern part of the basin.

(Figure 8)

(Figure 9)

The good agreement found in the Amazon basin can be attributed to the explicit representation of surface water reservoir
(channels and floodplains), which has been demonstrated to play an important role on both magnitude and timing of TWS (Alkama et al., 2010; Paiva et al., 2013; Getirana et al., 2017a). Other authors have pointed out that the contribution of surface storage to TWS is also potentially high in the Orinoco (~45 %) (e.g., Frappart et al., 2014), suggesting a large underestimation of the soil storage (in the eastern part of the basin) because anomalies of water level were reasonably well simulated. Indeed, surface storage has been understood as a major component of TWS variability over tropical regions of
South America, and may also be relevant for large rivers crossing semiarid areas such as the Sao Francisco (Getirana et al., 2017a). In the case of La Plata, the TWS amplitude is likely to be amplified if surface water is anticipated in time (Getirana et al., 2017a), which is probably occurring due to the low correlation of water levels previously simulated for the Paraguay River. In addition, the absence of a well-defined river system due to very flat terrains (e.g., Chaco Region, in the west part of La Plata) potentially favors the dominance of the groundwater dynamics over TWS, as already reported by Kuppel et al.,
(2015) in the Western Pampas more in the south.

It is worth mentioning that, for regions of South America located in mid latitudes, TWS is dominated by interannual variability rather than the seasonal cycle (Humphrey et al., 2016), where TWS amplitudes are generally lower and errors more apparent. Previous studies showed a strong negative trend in GRACE mascon solutions in the Colorado basin (e.g. Scanlon et al., 2016) that can be associated to a decrease in snow water equivalent over the dry Central Andes (Rivera et al.,
2017). Moreover, negative variations in glacier mass have been reported in southern Argentina/Chile over the Patagonia Icefields (Chen et al., 2007), which is probably the main responsible for the large RMSE observed at the extreme south of the continent. TWS in nearby semiarid areas is potentially affected by snow/glacier melting because the latter one is an important water source of Patagonian rivers flowing to the Atlantic Ocean (Pasquini and Depetris, 2007; Rivera et al. 2018). Inconsistencies along coastlines are also expected because of the smaller size of land mascons that increase uncertainty in
GRACE estimates (Wiese et al., 2016). Finally, in addition to issues related to model parameterization and depiction of hydrological processes (e.g., snowmelt), artificial reservoirs (dams) and lakes are not included in the current version of the South America MGB model, leading to additional uncertainties in TWS estimation.



## 4.2 Cross-scale comparison of river discharges from continental × global models

This section presents an assessment of MGB simulated discharges in comparison to the outputs from HTESSEL/CaMa-Flood, LISFLOOD and WaterGAP3 models, extracted from WRR-2 in the context of the eartH2Observe project (Schellekens et al., 2017; Martinez-de la Torre et al., 2018). This offers an interesting opportunity to evaluate if a continental model can provide better discharge estimates than global models in South America, as well as to identify the major shortcomings that should be addressed. To provide a concise spatial analysis, discharges from global models were reduced to their ensemble mean (Ensemble GM) and results are presented in terms of the difference of each metric (indicated by "d_metric"), i.e., by subtracting the performance of MGB from the performance of the Ensemble GM (Fig. 10). Bias and DI values are given in terms of absolute differences (d_Abs(metric)) to make both under- and overestimations comparable. Therefore, positive values indicate that MGB outperforms the ensemble mean of global models and vice-versa. Detailed performance metrics of each model can be found in Supplementary Material S3.

The continental model presents considerable improvements for all metrics over most of the South America regions when compared with the global ensemble mean. Overall agreement of simulated and observed discharges is much better (d_KGE > 0.8) over semiarid regions (e.g., East/Northeast Brazil and most part of Argentina), which are strongly impacted by bias in the Ensemble GM (d_Abs (BIAS) > 60 %). In tropical regions with marked seasonality and dry winter (e.g., upper Parana headwaters), differences in bias are lower (–10 % < d_Abs (BIAS) < 20 %), which indicates that KGE performance depends mainly on the variability of flows that is not captured by the Ensemble GM. Correlation is considerably higher over the Paraguay River (d_r > 0.4), highlighting the strong influence of hydrodynamic effects and complex processes in the Pantanal and Chaco regions, as documented by regional studies (e.g., Paz et al., 2011, 2014; Bravo et al., 2012, Pontes, 2016). There is also a clear correlation improvement in rivers such as the Araguaia, Amazonas and lower Magdalena, which are also affected by river–floodplain interactions with consequent flood peak attenuation (e.g., Paiva et al., 2013; Lininger and Latrubesse, 2016; Angarita et al., 2017; Pontes et al., 2017). A similar performance is observed for timing (d_Abs (DI)) with absolute differences being larger than 20 days, which also occur in the main stem of the Orinoco basin. In terms of NSE, the largest differences in performance previously observed for the KGE now extend to the main Amazon River, to its tributaries in the eastern region (i.e., Tapajos and Xingu) and also to both Magdalena and Tocantins–Araguaia basins, with values of d_NSE ≥ 0.8. With respect to low to medium flows (d_NSElog), there is a similar pattern to d_KGE (except for East Brazil), although with more pronounced differences in the Amazon and Magdalena regions.

The Ensemble GM performs relatively well in all statistics over temperate regions with the absence of lowland rivers (e.g., southern Brazil and South Chile), but for the entire continent, results can be considered better than MGB only for a limited number of gauge stations. For example, correlation is slightly reduced for the continental model (–0.1 < d_r < –0.2) in areas over the Parnaiba basin and Chile, while a marked decrease in timing performance (d_Abs (DI) < –20 days) is observed in dry Argentinian rivers like Salado (southwest of La Plata basin) and Desaguadero (Colorado basin). Poor estimates of river geometry and large overestimation of flows in these regions may be causing excessive flooded areas and consequent peak





attenuation. Regarding intermediate to low flows, considerable differences in model performance are observed mainly over specific rivers in East Brazil and parts of Amazon basin near the Andes Cordillera (d_NSElog < –0.8), as well as in regions over South Chile (–0.2 < d_NSElog < –0.6) that are potentially affected by snowmelt.

5 (Figure 10)

Table 3 shows differences in median discharge statistics for each global model and also for the Ensemble GM in comparison to the MGB continental model. Because LISFLOOD and WaterGAP3 account for reservoir impacts in their model structure, gauge stations with naturalized discharge data ($n = 98$) were excluded from the analysis to provide a fair assessment. Results 10 clearly show that MGB has a better overall performance in comparison to each of global models. Except for the Ensemble GM, differences between model performances (global × continental) are quite similar for KGE (~0.45) and NSElog (~0.5), while being highly variable for both NSE (~1 to ~1.8) and bias (~4 % to ~30 %). Differences in median DI are between 1 and 2 days, which can be important for cases where flood timing is around this order of magnitude. Among the estimates from global models only, the Ensemble GM outperforms four out of the six metrics analyzed (KGE, NSE, NSElog and DI) 15 with correlation (d_r = 0.03) equivalent to the best of global models for this metric (LISFLOOD, d_r = 0.02). A reduction in performance occurs only when bias is evaluated, where 50 % of the gauge stations have an absolute difference equal to or greater than 11 % compared with differences in HTESSEL/CaMa-Flood (d_Abs (BIAS) ≈ 8 %) and WaterGAP3 (d_Abs (BIAS) ≈ 4 %). In the assessment by Beck et al., (2017a) for basins < 10 000 km² around the world, LISFLOOD also had an advantage in correlation when compared with other global models, while WaterGAP3 demonstrated problems related to 20 baseflow index, which may be indicated here by the largest difference of NSElog (d_NSElog ≈ 0.6).

(Table 3)

The set of boxplots shown in Fig. 11 summarizes the individual performance of continental and global models. Results are 25 presented for some of the representative South America basins and also for the entire continental region, using a subset of metrics (KGE, NSE, BIAS and DI). In addition, to evaluate why the continental MGB model presents improved performance, a few degraded configurations were tested: hydrodynamic routing with calibrated rainfall-runoff parameters, i.e., the reference simulation (MGB_HD_calib), hydrodynamic routing with uncalibrated rainfall-runoff parameters (MGB_HD_noCalib), non-hydrodynamic routing with calibrated rainfall-runoff parameters (MGB_noHD_calib) and non-30 hydrodynamic routing with uncalibrated rainfall-runoff parameters (MGB_noHD_noCalib). For the uncalibrated MGB versions (noCalib), a single set of parameters was adopted corresponding to the median values resulting from model adjustment (as shown in sect. 3.3). It is important to note that only rainfall-runoff parameters were reduced to their median values, while river routing parameters (Manning coefficient and river geometries) remained unchanged.



(Figure 11)

Results indicate that global models have important limitations in representing daily discharges in South America. In absolute

terms, more than 40 % (60 %) of the gauge stations show negative or close to zero KGE (NSE) values. These models tend to overestimate discharges in the continent, with median bias ranging between +10 % and +50 %. In general, the performance among global models is variable according to the analyzed region and metric, which is supported by the large boxplot ranges. None of the models has a clear advantage with respect to all statistics, and this is especially valid for NSE and KGE. In the Amazon, KGE values present a more uniform pattern than in other regions with a median value close to 0.5, while

models agree in a reasonable number of positive KGE and NSE values. Performance in the La Plata basin is highly variable between models, and this is the only region in which both systematic underestimation (HTESSEL/CaMa-Flood with median BIAS ≈ –20 %) and overestimation (LISFLOOD with median BIAS ≈ +20 %) is observed. Moreover, performance of the global models in basins with semiarid regions (e.g., Sao Francisco and Parnaiba) is extremely poor for KGE and NSE (median < –1 and < –2 respectively), which are probably associated to a dramatic overestimation of flows in these regions.

WaterGAP3 shows a lower bias for all basins, but simulated peak flows occur too early according to DI for South America (percentile 25 % of DI ≈ –10 days). On the other hand, LISFLOOD appears to have a systematic delay in flow timing with more pronounced values over the Amazon (median DI ≈ +10 days), and also a strong wet bias. For instance, median values of bias in LISFLOOD are larger than 40 % for the entire continent and exceed 100 % in basins such as Sao Francisco and Parnaiba. Absolute DI values are generally lower for HTESSEL/CaMa-Flood (median DI closer to 0) and this model usually

shows an intermediate performance with respect to other metrics in comparison to LISFLOOD and WaterGAP3. Furthermore, the Ensemble GM shows a better overall performance when compared with each of these models alone, but still producing a similar number of negative KGE and NSE values (33 % and 60 % of the gauges, respectively).

Simulated discharges after setting MGB with a single set of rainfall-runoff parameters (MGB_noHD_noCalib) results in positive values for both KGE and NSE median values, varying between 0.3 and 0.6 for the entire continent. Apparently, the

uncalibrated version of MGB also outperforms global models in South America except for basins with semiarid regions (e.g., Sao Francisco and Parnaiba), where performances seem to be very dependent on parameter adjustment. The introduction of hydrodynamic routing (MGB_HD_noCalib) causes a slight improvement in NSE and KGE but this effect is more evident in the Amazon and especially over the La Plata basin (percentile 25 % of NSE changes from –1.5 to 0). Improvements in flow timing (DI) for both Amazon and La Plata are also observed after including the HD routing method, although excessive

delays occur in Sao Francisco and Parnaiba because of the large bias that leads to an excess of floodplain attenuation (see MGB_HD_noCalib × MGB_HD_Calib). Furthermore, boxplot ranges are considerably smaller for KGE, NSE and bias in the default MGB simulation (MGB_HD_Calib) with respect to global models, and this reduction can be mostly seen in both MGB calibrated versions.





Our results are in agreement with other studies from the literature, which highlight the large influence of model structure and parameterization in addition to meteorological forcing (e.g., Haddeland et al., 2011; Gudmunsson et al., 2012; Zhou et al., 2012; Beck et al., 2017a). Regarding global models, other studies also stress the large number of negative NSE values resulting from LSMs and GHMs in many basins around the world (e.g., Zhang et al., 2016; Beck et al., 2017a), and it was

even evidenced in South America (e.g., Wu et al., 2014). In particular, WaterGAP3 is expected to produce a lower bias because it is calibrated in terms of mean annual flow (Döll et al., 2009; Müller Schmied et al., 2014), and the systematic advance in timing is probably caused by the simple variable velocity equation (based on Manning) used for computing flow routing. In the case of LISFLOOD, large overestimation of flows in comparison to other global models has already been reported in the context of the eartH2Observe project (Beck et al., 2017a), where it showed the lowest estimate of potential

evaporation (Schellekens et al., 2017). Indeed, this excessive wet bias is one of the possible reasons for the observed delay in flows, but this is not the only factor because large overestimations are concomitantly found in regions where DI is negative. This is the case of eastern Amazon tributaries located downstream of Obidos, such as the Tapajos and Xingu (see Supplementary Material S3). The interplay between Manning coefficient and groundwater parameters and their influence on flow timing of LISFLOOD has been shown in recent studies (e.g., Revilla-Romero et al., 2015; Zajac et al., 2017), and here

may compensate for limitations of the double-kinematic wave (channel + floodplain) used for river routing, especially in the Amazon. This suggests that calibrating large basins with lowland river systems using few downstream stations (such as Obidos) should be taken with care if hydrodynamic routing is not accounted for in the model structure.

Although some authors pointed to a clear (and general) underestimation of HTESSEL (e.g., Haddeland et al., 2011; Beck et al., 2017a), our results showed that it occurs only in the La Plata Basin but not in other regions, which may be related to the

precipitation forcing used in WRR-2 (MSWEP). With respect to model performances, the relatively better flow timing of HTESSEL can be attributed to CaMa-Flood routing, but the advantages of this coupling were below the expected ones when looking at other statistics. It is worth mentioning that default parameters for river routing were used within WRR-2 simulations, i.e., the CaMa-Flood model was not calibrated (Martinez-de la Torre et al., 2018). This could be one of the reasons that low values of NSE are found over the Amazon main stem for this model (see Supplementary Material S3).

Characteristics such as timing and magnitude of flood waves in the hydrodynamic routing are very sensitive to channel geometry and roughness (Yamazaki et al., 2011; Paiva et al., 2013), but also to DEM vegetation effects (Baugh et al., 2013) that can impact the subgrid floodplain profiles (Paiva et al., 2011; Yamazaki et al., 2011). Furthermore, Zhao et al., (2017) emphasize that the benefit of CaMa-Flood highly depends on the runoff fields simulated by the coupled LSM. Our results showed that discharge estimates of an uncalibrated model are improved over Amazon and (mainly) La Plata basins after

inclusion of hydrodynamic routing (MGB_HD_noCalib), provided that channel geometry and floodplain topography are reasonably well estimated. Apart from the particular issues of WaterGAP3, LISFLOOD and HTESSEL/CaMa-Flood over South America, our findings reinforce the conclusion of other authors who recommend the ensemble mean of global models as the most reliable estimate (e.g., Haddeland et al., 2011; Gudmunsson et al., 2012; Schellekens et al., 2017; Hattermann et al., 2018), and it occurs even when discharges of a small number of models are averaged (three in the present case).



As outlined by many studies (e.g., Haddeland et al., 2011; Gudmundsson et al., 2012; Pechlivanidis and Arheimer, 2015; Zhang et al., 2016; Krysanova et al., 2017), performance of both regional and global models generally reduces when there is a transition from wet to dry conditions. In semiarid regions, satellites have several limitations in capturing rainfall intensities due to the local, convective nature of the precipitation, and they often overestimate the occurrence of rainfall because

raindrops are likely to evaporate (i.e sub-cloud evaporation) before reaching the surface (Dinku et al., 2010; Sunilkumar et al., 2015; Beck et al., 2017c). In addition, runoff generation mechanisms are strongly nonlinear and depend too much on storage processes, which are parameterized with large uncertainty (Gudmundsson et al., 2012). For instance, there is little knowledge about the influence of transmission losses, their partitioning between its main components (e.g., infiltration/evaporation from channels or floodplains) (Jarihani et al., 2015) and the dominant mechanisms of losing/gaining

water according to different periods of the wet season (Costa et al., 2013). Processes such as reinfiltration of surface runoff, lateral redistribution of subsurface runoff and hydraulic-connected stream-aquifer interactions have been shown to be necessary for hydrological modeling in Northeast Brazil (Güntner and Bronstert, 2004; Costa et al., 2012, 2013), but are not explicitly accounted for in any structure of the assessed models. Therefore, a systematic underestimation of continental ET and consequent overestimation of flows is expected in dry regions (e.g., Alkama et al., 2010; Haddeland et al., 2011; Zhang

et al., 2016). Uncertainties about human interferences in water resources (i.e., small ponds and reservoirs, water abstractions) may also play an important role (Hanasaki et al., 2018), especially in regions where data are scarce. Nevertheless, the complexity of the global models assessed herein makes it difficult to explain the real factors that impact discharge estimates.

## 6 Summary and conclusions

For the first time, a regional-scale, fully coupled hydrologic–hydrodynamic model (MGB) was applied to a continental

domain (South America). Model results were assessed using observed discharges and water levels from both in situ/satellite altimetry at an unprecedented gauge network over the continent, together with estimates of TWS and ET from remote sensing and other data sources. In addition, a cross-scale assessment (i.e., regional/continental × global models, the latter acquired from the eartH2Observe project) was conducted with the novelty of using spatially consistent, daily discharge data for model comparison.

Regarding continental modeling, analyses showed a satisfactory agreement between simulated and observed discharges, with NSE (KGE) > 0.6 in more than 55 % (70 %) of the gauges. The performance was generally better in large rivers and humid regions, but worse in areas with semiarid to arid climates, influence of snowmelt or draining complex orography such as the Andes Cordillera. Similar results were found for water levels, despite having 50 % of the gauges with large under- and overestimation of amplitudes (> |30 %|). The model was able to capture patterns of seasonality and magnitude of ET and

TWS in many parts of the continent, especially when results were averaged to the scale of large South American basins (e.g., Amazon, La Plata, Orinoco, Tocantins–Araguaia, Sao Francisco, Parnaiba). In addition, model errors in simulating discharges were also found in other hydrological variables, which demonstrate the importance of assessing model results





using multiple data sources. Uncertainties were attributed to deficiencies in process representation and simplifications in parameterization, as well as to limitations of the datasets used as model input and validation.

We found that continental-scale modeling significantly improves discharge estimates compared with individual global models (i.e., HTESSEL/CaMa-Flood, LISFLOOD and WaterGAP3). Differences in performance reached median values around 0.45 for both KGE and NSElog, being larger than unity for NSE. Global models were marked by a large number of negative NSE values (> 60 % of streamflow gauges) and resulted in highly variable performances when evaluated over multiple gauges within large basins. In general, these models were affected by positive bias mainly in East/Northeast Brazil and regions over Argentina, as well as in San Francisco, Parnaiba and Magdalena basins. Timing errors were predominantly found in rivers with floodplain effects, such as the Amazon, La Plata, Tocantins–Araguaia, Orinoco and lower Magdalena. Nevertheless, global models demonstrated a good ability to predict daily discharges over temperate, humid regions with the absence of lowland rivers (e.g., southern Brazil and south Chile), while performing reasonably in the Amazon basin. An ensemble mean generated by averaging discharges from HTESSEL/CaMa-Flood, LISFLOOD and WaterGAP3 resulted in a better (overall) estimate than any of these models alone, especially when benchmarked against the continental model.

As global models assessed in this study were originally run with grid resolutions varying between 0.08° and 0.25°, additional questions arise of how much discharges can be improved in South America just by increasing their spatial resolution. Our analyses clearly showed that model calibration and hydrodynamic routing have a major impact on simulation of daily discharges in this continent. Calibration was found to be important in most regions but mainly in drier basins (e.g., Parnaiba and Sao Francisco), where models generally fail to represent the underlying hydrological processes. In addition, a hydrodynamic module is essential to achieve a suitable representation of both magnitude and timing in major river systems, especially in cases where flows are dramatically attenuated by floodplains (e.g., the Paraguay River). If channel geometry is reasonably well estimated, increasing only the physical description of river routing may improve hydrological simulations over Amazon and La Plata basins, but the expected benefit of this coupling occurs when calibration of rainfall-runoff parameters is performed together. This must be conducted in a spatially consistent way, i.e., not considering only a single downstream gauge of large basins to reduce potential issues related to parameter compensation. Notwithstanding the efforts required by a manual calibration, as performed in this study, expert knowledge of local modelers allows better exploration of the available data while respecting model limitations, as optimization techniques are too much linked to objective functions.

Among the largest basins in South America, challenges still remain in simulating the La Plata (in particular, the Paraguay River) and this is a major concern given its economic importance for many countries of the continent. In this case, improvements in representing discharges are expected after conducting a more detailed river discretization in the Pantanal region and inclusion of a quasi-2D connection scheme over its floodplain (see Fleischmann et al., 2018). Using our online approach (i.e., fully coupled hydrologic–hydrodynamic) together with routing water in multiple downstream directions will enable both representing diffuse flows over the floodplain and feedback between surface water and soil processes, which can be very pronounced over large, seasonally flooded tropical wetlands such as the Pantanal (Paz et al., 2011, 2014).





Finally, the results found in this study confirm that MGB can be applied to South America as an alternative to global models. This underscores the importance of past regional modeling experiences, which can help to overcome limitations of global discharge estimates at the continental scale. We hope that moving from regional toward continental hydrologic–hydrodynamic modeling will bring new opportunities for operational practices such as real-time hydrological forecasting, which is the topic of an ongoing research. Nevertheless, several improvements should be carried out in the model structure not only to achieve a better understanding of the underlying processes but also to provide further insights about human impacts on South American water resources. This includes the representation of reservoirs, lakes and water abstractions. Uncertainties in model parameters are also important to be addressed and should be further investigated.

*Data availability*. Results from the MGB model are available upon request to the corresponding author. All other datasets used in the present study can be accessed using the websites cited in this manuscript.

*Author Contributions*. V. A. S. worked on the MGB model for South America, designed the study, performed model calibration/simulations, managed all datasets (in situ, remote sensing and global models) and produced the manuscript; R. C. D. P. helped on study design, MGB code optimization, interpretation of results, contributed to writing of the manuscript and conducted paper review; A. S. F. and F. M. F. assisted on MGB calibration and conducted paper review; A. L. R. helped with the interpretation of results. P. R. M. P. provided support with GIS routines and MGB simulations over La Plata basin; A. P. and S. C. provided support with satellite altimetry data and associated analyses; W. C. was the research supervisor, contributed to writing of the manuscript and conducted paper review.

*Competing interests*. The authors declare that they have no conflict of interests.

*Acknowledgments*. The first author would like to acknowledge the Brazilian National Council for Scientific Research (CNPq) for the financial support (Project: "South America Flood Awareness System – SAFAS", under grant number 422422/2016–9); the institutions ANA, SENAMHIs, IDEAM, DGA, INA, HyBAM, GRDC and ONS for providing in situ discharge and water stage data; the LEGOS–HydroWeb for both Envisat and Jason-2 satellite altimetry data; the Jet Propulsion Laboratory (JPL) for GRACE mascon data; Princeton University for the CDR–ET data; the eartH2Observe project for WRR-2 discharges from global models (WaterGAP3, LISFLOOD and HTESSEL/CaMa-Flood) and, in particular, Martina Floerke, Gabriel Fink, Ad de Roo, Emanuel Dutra and Gianpaolo Balsamo for their kind support with the global models used herein.




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



**Table 1. Parameters used to calibrate the rainfall-runoff module of MGB, including their respective (typical) range of values.**

| Parameter | Description | Unit | Min | Max |
|---|---|---|---|---|
| Wm | Maximum water storage | mm | 50 | 2000 |
| b | Controls the distribution of water storage capacity of the soil | - | 0.01 | 1.6 |
| Kbas | Percolation rate from soil to groundwater | mm $\Delta t^{-1}$ | 0.1 | 4 |
| Kint | Saturated soil hydraulic conductivity | mm $\Delta t^{-1}$ | 4 | 40 |
| Kinf | Infiltration rate from floodplains when soil is completely dry | mm $\Delta t^{-1}$ | - | |
| XL | Soil porosity index | - | Default = 0.67 | |
| Cb | groundwater reservoir residence time | h | 800 | 8000 |
| Ci | Adjustment factor for subsurface reservoir residence time | - | 50 | 200 |
| Cs | Adjustment factor for superficial reservoir residence time | - | 1 | 30 |





**Table 2. Efficiency metrics used in this study.**

| Metric | Abbreviation | Assessment | Variables | Equation |
|---|---|---|---|---|
| Pearson's Correlation Coefficient | r | Linear correlation | Discharge, Water level, TWS | $\dfrac{\sum[(x_{sim}-\mu_{sim})(x_{obs}-\mu_{obs})]}{\sqrt{\sum(x_{sim}-\mu_{sim})^2}\sqrt{\sum(x_{obs}-\mu_{obs})^2}}$ |
| Modified Kling–Gupta Efficiency | KGE | Overall Performance | Discharge | $1-\sqrt{(1-r)^2+\left(1-\dfrac{\mu_{sim}}{\mu_{obs}}\right)^2+\left(1-\dfrac{CV_{sim}}{CV_{obs}}\right)^2}$ |
| Nash–Sutcliffe Efficiency | NSE | High Flows | Discharge, Water level | $1-\dfrac{\sum(x_{sim}-x_{obs})^2}{\sum(x_{obs}-\mu_{obs})^2}$ |
| Log-transformed Nash–Sutcliffe Efficiency | NSElog | Low Flows | Discharge | $1-\dfrac{\sum[Log(x_{sim})-Log(x_{obs})]^2}{\sum[Log(x_{obs})-Log(\mu_{obs})]^2}$ |
| Overall BIAS (%) | BIAS | Under- and overestimation (volume) | Discharge | $\left(\dfrac{\sum x_{sim}-\sum x_{obs}}{\sum x_{obs}}\right).100$ |
| BIAS in standard deviation (%) | σ BIAS | Under- and overestimation (anomalies) | Water levels, TWS | $\left(\dfrac{\sigma_{sim}-\sigma_{obs}}{\sigma_{obs}}\right).100$ |
| Delay Index (days) | DI | Timing errors | Discharge | $\max r_{xy}[x_{sim};x_{obs}],\quad lag\ [-100;+100]$ |
| Root Mean Square Error | RMSE | Deviation of predicted values | ET, TWS | $\sqrt{\dfrac{\sum(x_{sim}-x_{obs})^2}{n}}$ |

Where: $x_{sim}$ = simulated variable; $x_{obs}$ = observed variable; $\mu_{sim}$ = mean of simulated variable; $\mu_{obs}$ = mean of observed variable; CV = coefficient of variation, equal to $\sigma/\mu$; $r_{xy}$ = cross-correlation



**Table 3. Median values of discharge metrics for South America, computed as the performance difference between continental and global models. Lower values show better performance for a given global model when benchmarked against the MGB continental hydrologic–hydrodynamic model. Gauge stations with naturalized flows were removed from the analysis to provide a fair comparison.**

| Model difference | d_r | d_KGE | d_NSE | d_NSElog | d_Abs(BIAS) [%] | d_Abs(DI) [days] |
|---|---|---|---|---|---|---|
| MGB – HTESSEL/CaMa-Flood | 0.11 | 0.48 | 1.35 | 0.53 | 8.3 | 1 |
| MGB – LISFLOOD | 0.02 | 0.44 | 1.86 | 0.48 | 32.5 | 1.5 |
| MGB – WaterGAP3 | 0.16 | 0.44 | 1.05 | 0.57 | 3.8 | 2 |
| MGB – Ensemble GM | 0.03 | 0.26 | 0.91 | 0.27 | 11.0 | 0 |





**Figure 1: South America maps showing: (a)Countries and major hydrological regions according to FAO and ANA classifications, (b)Major wetlands and lowland regions, adapted from Lehner and Döll (2004), (c)Mean Annual Precipitation derived from MSWEP dataset (Beck et al., 2017b), (d)Köppen–Geiger updated climate classification from Kottek et al., (2006), (e)Relief map based on the Bare-Earth SRTM (O'Loughlin et al., 2016), including main rivers and (f)Mean annual flow at discharge gauge stations used in this study.**





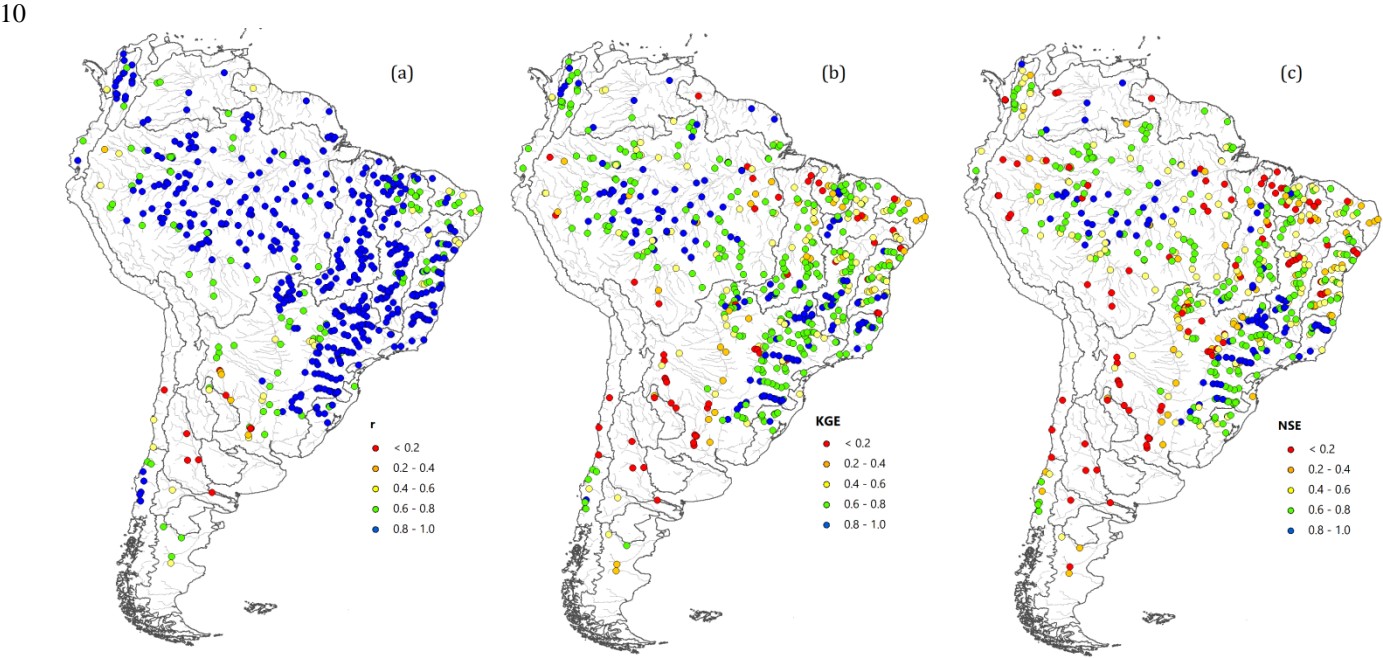

**Figure 2: Discharge performance over South America in terms of (a)Correlation (*r*), (b)Kling–Gupta Efficiency (KGE) and (c)Nash–Sutcliffe (NSE).**





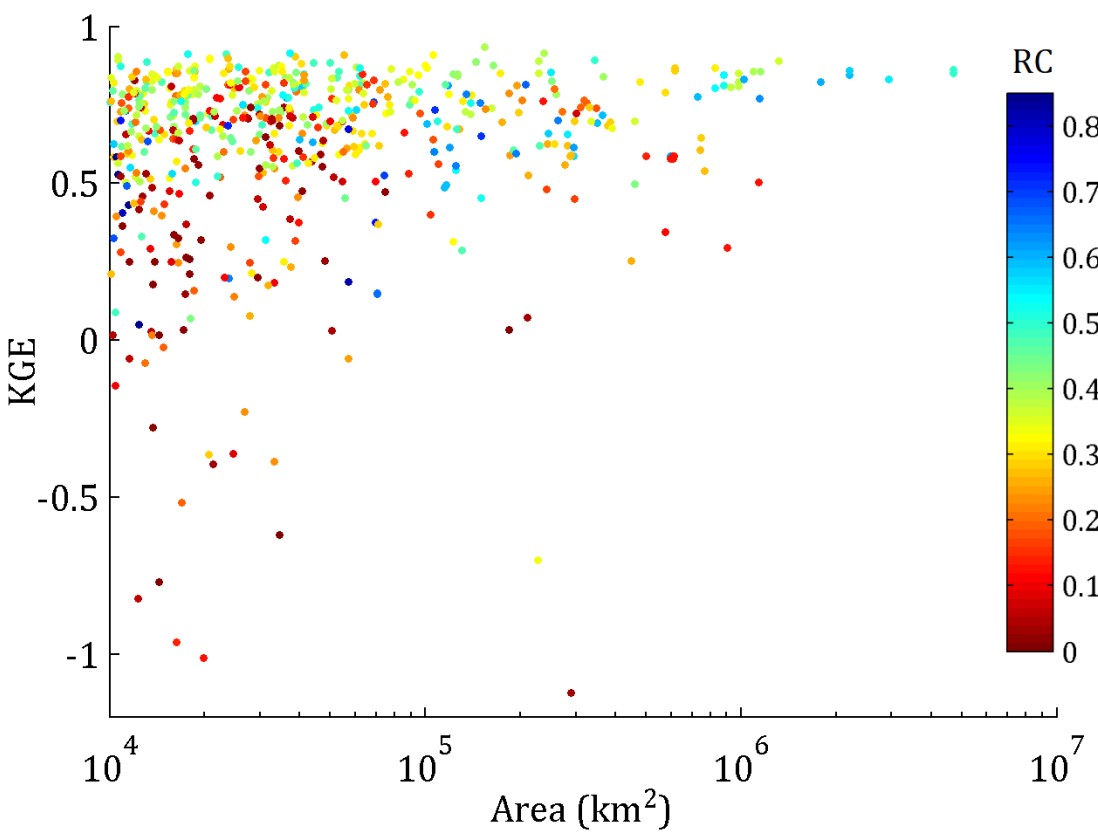

**Figure 3: MGB model performance (KGE) versus RC and drainage area over South America.**







**Figure 4: Comparison between observed (black) and simulated discharges for major South American rivers. Model results are shown considering both hydrodynamic (HD; red color) and non-hydrodynamic (noHD; gray color) routing methods.**





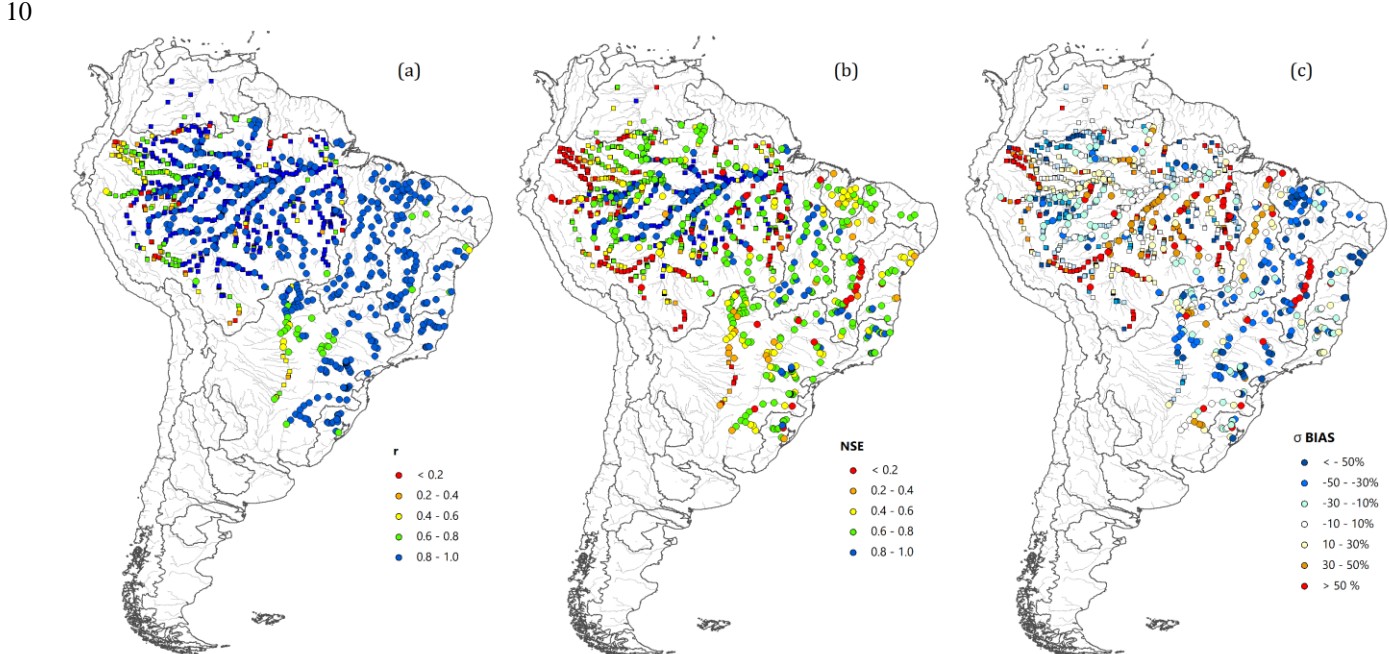

**Figure 5: MGB performance for simulated water levels over South America in terms of (a)Correlation (*r*), (b)Nash–Sutcliffe (NSE) and (c)Bias in standard deviation (σBIAS). In situ and satellite altimetry locations are shown in circle and square symbols, respectively.**





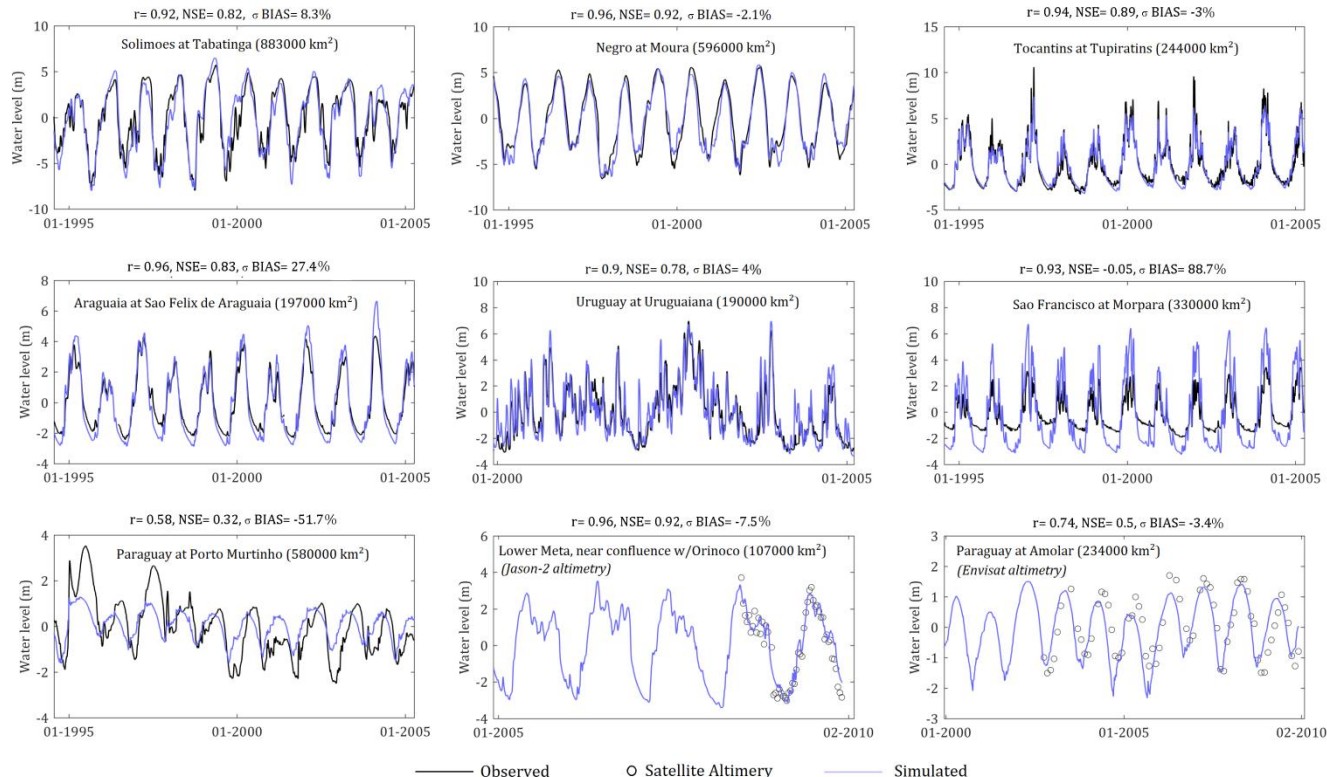

**Figure 6: Comparison between simulated (blue) and observed (black) water level anomalies in major South American rivers for in situ gauges (continuous lines) and satellite altimetry (circles) at virtual stations (VS).**







**Figure 7: Comparison between MGB and CDR ET estimates in terms of RMSE (left) and seasonality for major South American basins (right). The light gray area represents the proxy of the CDR uncertainty, i.e., the mean deviation of all datasets (within CDR) from the ensemble mean (Zhang et al. 2018).**





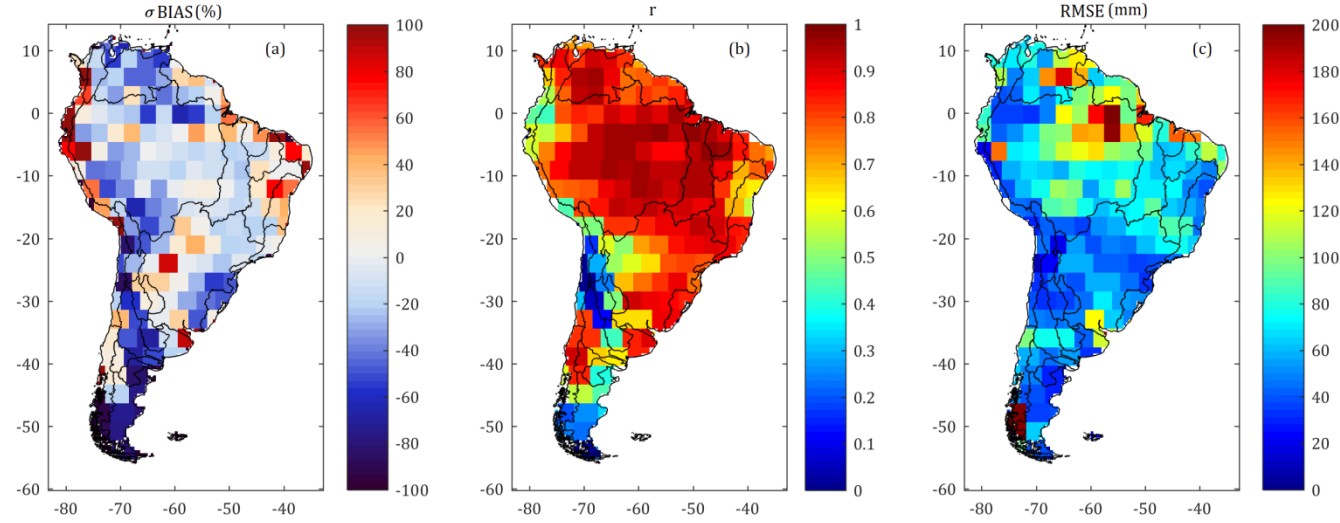

Figure 8: Comparison between MGB and GRACE (JPL RL05 v2 mascon solution) TWS anomalies in terms of (a)Bias in standard deviation, (b)Correlation and (c)RMSE.





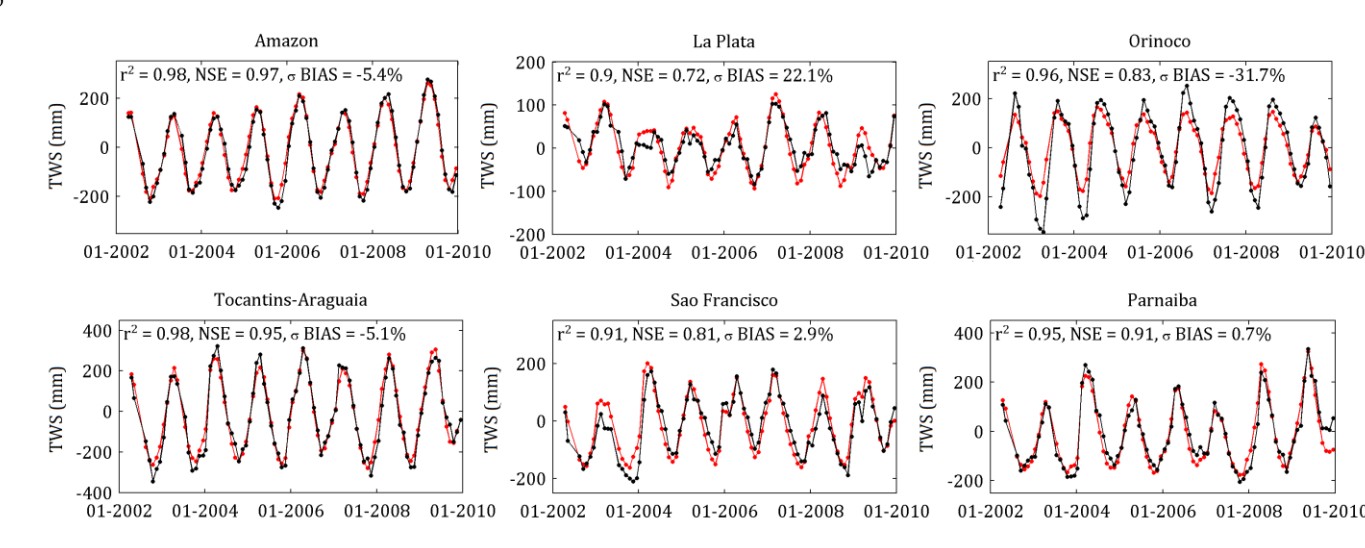

**Figure 9: Comparison between MGB and GRACE (JPL RL05 v2 mascon solution) monthly TWS anomalies for major South American basins.**




**Figure 10: Difference between performances of MGB and the Ensemble GM for discharge metrics. Values considered as not**
**significant (gray) are within the ranges −0.05 to +0.05 (d_r, d_KGE, d_NSE and d_NSElog), −5 % to 5 % (d_Abs(BIAS)) and −2 to**
**+2 days (d_Abs(DI)).**



**Figure 11: Boxplots of global (above center line) and continental (below center line) model performances for different South American regions. MGB model configurations: hydrodynamic routing with calibrated rainfall-runoff parameters (MGB_HD_calib), hydrodynamic routing with uncalibrated rainfall-runoff parameters (MGB_HD_noCalib), non-hydrodynamic routing with calibrated rainfall-runoff parameters (MGB_noHD_calib) and non-hydrodynamic routing with uncalibrated rainfall-runoff parameters (MGB_noHD_noCalib). Gauge stations with naturalized flows were removed from the analysis to provide a fair comparison.**