# Peer review of "Toward continental hydrologic-hydrodynamic modeling in South America"

_Hydrology and Earth System Sciences, 2018_

## Referee Comment (RC1) · Anonymous Referee #1 · 12 Jun 2018

This m/s presents a continental hydrological model for South America. It is forced by global rainfall and climate data and calibrated to streamflow records for a large number of stations. The agreement with recorded streamflow is presented, as well as that with satellite-derived evaporation (ET) and total water storage (TWS). The agreement with observed streamflow is better than that of an ensemble of 3 global models driven by the same precipitation estimates.

Overall assessment: This appears an overall competent and sound study, but I am missing some truly new scientific insights. The abstract suggests the main insights are (1) calibrating rainfall-runoff parameters is necessary to simulate discharge appropriately; and (2) implementing hydrodynamic routing is also important. I don't think either of those is really very novel. I do not think there was ever any doubt that parameter calibration against streamflow records was going to improve the agreement with those same records (noting that the "appropriately" used in the abstract is obviously a subjective term, or at least one that would have to be purpose-specific). The second conclusion also hardly seems surprising and has been shown in previous studies, specifically for the Amazon basin. Indeed, the authors provide several literature references that offered those very conclusions.

On the positive side, I do think this is an interesting study that has the potential to be a valuable contribution. I thought some of the most interesting contributions from this study were:

1) There is a much larger set of streamflow gauging stations in existence in South America than is represented in global databases and typically used to calibrate global models;

2) The use of a large number of altimetry-derived water level records is interesting; and

3) The authors provide some interesting commentary on the hydrological conditions that likely explain consistently poor performance by global models in some of the basins in South America.

I was somewhat surprised that the majority of forcing and spatial parameterisation approaches used for the "regional" MGB model were, in fact, the same as used for global models. Furthermore, Fig. 11 appears to suggest that the inclusion of hydrodynamic routing was a minor factor in explaining the generally better performance. Therefore, it would seem that the larger number of streamflow gauges and their good use in a more intensive calibration were the real reasons for better performance. That in its own right is useful, as it sets a benchmark that global models should be able to achieve with appropriate parameter calibration.

What is unclear, however, is whether that would go at the detriment of the agreement with other observations of the water cycle, such as ET and TWS. It is common

that a heavy emphasis on streamflow calibration leads to deterioration in other terms. Therefore, I was surprised that the authors did not include the global models in their comparison against ET and TWS, to assess whether those were simulated better or worse. (The altimetry water levels are less relevant in such a comparison, as one would assume that better discharge simulation also produces better water level simulation. Nonetheless, a comparison with the global models might still have been of interest.)

In summary, the present m/s mainly seems to assert the common "our model is better than theirs", which is not very insightful as it appears almost entirely due to calibration. There are however some good opportunities to make this a more valuable (and cited!) contribution:

1) Propose these model simulations, along with the station and altimetry records, as a benchmark for global models by making them directly available online to the global modelling community. To make the MGB model acceptable as a benchmark for an allround hydrological model, you should demonstrate whether the global models are also less effective in simulating ET and TWS. This would provide insight into whether only the streamflow simulations can be considered benchmark, or the other water cycle components as well.

2) Provide more discussion and emphasis on the understanding of the hydrological conditions of some of the "problem" basins, so that they might become a valuable "stress test" of hydrological model performance.

---

## Referee Comment (RC2) · G. J.-P. Schumann (Referee) · 12 Jun 2018

This paper describes in detail a much needed continental-scale cross-comparison study at the continental scale of the typical regional MGB model using not only different global scale models but also observed or observation inferred variables (e.g. TWS from GRACE, satellite altimetry). The paper is technically very sound and strong and I did not see any problems with the methods employed.

I really enjoyed reading this paper and although it is fairly long in places, I think it is written in a very comprehensive way and very well organized and presented - I applaud such work and writing. Well done! This said, there are some main points I would like to highlight and see addressed before publication.

[Figure]

- I think it would benefit the paper a lot by listing a number of steps or recommendations to follow for large-scale hydrologic model assessment or validation

- It looks to me as though generally speaking the headwaters are difficult to get right or better said "to agree with other models", which means to me that they are generally very difficult to model correctly. This is of course not surprising given that the topographic complexity and hydrological processes in these regions are not well represented in the models. It would be useful if the authors could comprehensively outline the reasons for those "problem areas".

- As far as I understand the authors, model calibration is still challenging and therefore could also be responsible for explaining some or even most of the differences observed between different models. Logically it follows that there should be the general recommendation to define and build a set of data that should be used for calibration of large scale models, so that comparison studies later are even more valuable. I think the authors, if they can agree, should call for such a data set in their section of "Model adjustment" (section 3.3) or later in the conclusion is maybe even a better place.

---

## Author Comment (AC1) · 5 Jul 2018

**Text in bold: Reviewer comments**; *Text in italic: Authors response*

**Anonymous Referee #1**

**This m/s presents a continental hydrological model for South America. It is forced by global rainfall and climate data and calibrated to streamflow records for a large number of stations. The agreement with recorded streamflow is presented, as well as that with satellite-derived evaporation (ET) and total water storage (TWS). The agreement with observed streamflow is better than that of an ensemble of 3 global models driven by the same precipitation estimates.**

**Overall assessment: This appears an overall competent and sound study, but I am missing some truly new scientific insights. The abstract suggests the main insights are (1) calibrating rainfall-runoff parameters is necessary to simulate discharge appropriately; and (2) implementing hydrodynamic routing is also important. I don't think either of those is really very novel. I do not think there was ever any doubt that parameter calibration against streamflow records was going to improve the agreement with those same records (noting that the "appropriately" used in the abstract is obviously a subjective term, or at least one that would have to be purpose-specific). The second conclusion also hardly seems surprising and has been shown in previous studies, specifically for the Amazon basin. Indeed, the authors provide several literature references that offered those very conclusions.**

*We are grateful to the reviewer for reading our manuscript and pointing out relevant questions that need clarification.*

*To the best of our knowledge, this is the first study that provides a comprehensive cross-scale comparison between regional/continental x global models, supported by spatially distributed, daily discharge data. As stated by the reviewer, there is no doubt that calibration against discharge records leads to improvements in this same variable. Intercomparison studies indicate that models with some degree of parameterization generally perform better on average (e.g., Zhang et al. al., 2016, Beck et al., 2017). On the other hand, results from this paper put light on what extent both calibration and improvement in model physics (routing) are expected to improve estimates of daily river discharges, focusing on different aspects (overall agreement, high and low flows, timing, bias) and regions of South America. To our knowledge, no other study provides such discussion.*

*Although these results do not necessarily show the performance limit that can be achieved by current global forcing data, we understand that these results are of interest to the modeling community that has been seeking for locally relevant hydrological estimates, especially in under development regions. Recent studies call for cooperation between scales (e.g., Archfield et al., 2015) and this work is going exactly on that direction. In summary, continental-scale modeling shows that it is possible to get better discharge estimates by using global data and methods that are currently available, as well as knowledge and methods developed for the studied region. Therefore, we believe*

*that our manuscript brings new information that is relevant in the context of regional, continental and global-scale hydrological modeling.*

*We agree with the reviewer that the term "appropriately" in the abstract is a subjective term. We also understand that some modifications in text are necessary to clarify the aforementioned statements.*

**On the positive side, I do think this is an interesting study that has the potential to be a valuable contribution. I thought some of the most interesting contributions from this study were:**

1) **There is a much larger set of streamflow gauging stations in existence in South America than is represented in global databases and typically used to calibrate global models;**

*Thanks for this comment. Yes, much of the available data are hardly accessed by modeling studies with global coverage. As shown in section 3.2.1, we have only selected gauge stations with drainage areas above 10000 km², that is, much data is still available to be used in the future for more detailed evaluations. It is worth mentioning that a lot of time is needed for data acquisition, quality check (time series, gauge location, drainage area) and standardization considering different institutions, and these efforts can put constraints on data usage. But we understand that handling local information is important since global databases sometimes suffer from "disinformative data" (Kauffeldt et al., 2013).*

2) **The use of a large number of altimetry-derived water level records is interesting;**

*Thanks for this comment. Indeed, the use of satellite altimetry is interesting to evaluate the performance of the routing component (in addition to ET and TWS that are used to evaluate the water balance), especially in regions where data are scarce.*

3) **The authors provide some interesting commentary on the hydrological conditions that likely explain consistently poor performance by global models in some of the basins in South America.**

*Thank you for taking a positive look at this discussion in the manuscript. This point, for example, comes right into the idea of reducing the gap between global and regional modeling. Studies focused on the global scale are usually not concerned with specific processes occurring in specific regions, but there are key studies - many of them on a regional scale - that provide further insights about modeling results and can partially explain the performance obtained. Therefore, we believe that studies at the continental scale can serve as a link for communicating relevant findings to a broader audience, helping to improve the general understanding.*

I was somewhat surprised that the majority of forcing and spatial parameterisation approaches used for the "regional" MGB model were, in fact, the same as used for global models. Furthermore, Fig. 11 appears to suggest that the inclusion of hydrodynamic routing was a minor factor in explaining the generally better performance. Therefore, it would seem that the larger number of streamflow gauges and their good use in a more intensive calibration were the real reasons for better performance. That in its own right is useful, as it sets a benchmark that global models should be able to achieve with appropriate parameter calibration.

What is unclear, however, is whether that would go at the detriment of the agreement with other observations of the water cycle, such as ET and TWS. It is common that a heavy emphasis on streamflow calibration leads to deterioration in other terms. Therefore, I was surprised that the authors did not include the global models in their comparison against ET and TWS, to assess whether those were simulated better or worse. (The altimetry water levels are less relevant in such a comparison, as one would assume that better discharge simulation also produces better water level simulation. Nonetheless, a comparison with the global models might still have been of interest.)

*Thanks for the comment. Yes, many of the techniques and databases used to prepare the continental model are similar to those used by global models. Such strategy indicates that the latter can achieve similar results even without a significant increase in the number of computational elements. As pointed out by the reviewer, parameterization has a greater impact on model results in comparison to improved routing, although efforts in calibration may be not effective in regions such as the Paraguay basin. Another problem is when a simplified routing structure is applied together with calibration focused in gauge stations with large drainage areas, especially in basins affected by floodplains (e.g., Amazon at Obidos). This can cause problems on flow timing at upstream regions as discussed on the manuscript.*

*Regarding the comparison between the continental x global models using variables such as ET and TWS the reviewer makes an interesting suggestion, but we understand that this would not fit into the context of the present study. We have focused on river discharge because this variable is widely used for water resources planning and practical applications (e.g., hydrological forecasting, reservoir operation). It would be difficult to demonstrate if a given model has better accuracy than another with respect to ET since the reference (CDR) has large uncertainties. In addition, to our knowledge, TWS for WRR-2 is not directly available for each model, but rather for an ensemble mean generated with the inclusion of several other global models (i.e., not only those used in this study). Yet, even if there is some interest in comparing water level anomalies, this would not be possible because water level is not an output of WRR-2.*

*We agree that calibration against discharge records could lead to decreased performance in other hydrological variables such as ET and TWS. This is the reason*

*why we performed several evaluations of both variables seeking to document model errors and to understand potential sources of uncertainty.*

**In summary, the present m/s mainly seems to assert the common "our model is better than theirs", which is not very insightful as it appears almost entirely due to calibration. There are however some good opportunities to make this a more valuable (and cited!) contribution:**

*Thanks for this comment. We will review the manuscript to make sure that analyses and discussions are constructive and not biased.*

*Model intercomparison studies sometimes make direct comparisons of performance (e.g., Xia et al., 2012, Zhang et al., 2016, Beck et al., 2016, Beck et al. 2017), and we understand that this is important because identifying model shortcomings is essential for future improvements to be made. In this context, there is no best model (first because it depends on the objectives) but rather issues of scale and the best use of available information. For instance, it is expected that models with a regional domain (even if forced with global data) will have, on average, better performance when compared with a continental model. Of course, it would be of high interest to understand which areas/rivers, flow conditions (high and low flows, bias, timing...) and for what reasons this happens, which definitely represents a reduction of the gap between different scales. The reviewer seems to agree with this idea when he/she states: "The authors provide some interesting commentary on the hydrological conditions that are likely to explain consistently poor performance by global models in some of the basins in South America."*

**1) Propose these model simulations, along with the station and altimetry records, as a benchmark for global models by making them directly available online to the global modelling community. To make the MGB model acceptable as a benchmark for an allround hydrological model, you should demonstrate whether the global models are also less effective in simulating ET and TWS. This would provide insight into whether only the streamflow simulations can be considered benchmark, or the other water cycle components as well.**

*Thanks for the suggestion. Indeed, model outputs will be available for public access in a specific website or by request to the authors. This will be indicated in the item "data availability" in the revised version of the manuscript. In addition, we can make available a list of the gauge stations in the supplementary material (name, coordinates, drainage area, respective institutions), since these data can be downloaded according to section 3.2.1. We can also recommend the development of a dataset to facilitate both model validation and intercomparison in South America, as suggested by Reviewer #2.*

*With respect to ET and TWS, we have previously indicated the reasons why it would not be feasible to make such a comparison at this moment, apart from the fact that the current study is already long.*

**Provide more discussion and emphasis on the understanding of the hydrological conditions of some of the "problem" basins, so that they might become a valuable "stress test" of hydrological model performance.**

*Thanks for the suggestion. We will add discussion in the revised version of the manuscript. This will certainly bring contributions for proposing ways of improving future modeling applications (both regional and global).*

*References*

*Archfield, S.; Clark, M.; Arheimer, B; Hay, L. E.; McMillan, H.; Kiang, J. E.; Seibert, J; Hakala, K; Bock, A.; Wagener, T.; Farmer, W. H.; Andréassian, V.; Attinger, S.; Viglione, A.; Knight, R.; Markstrom, S.; Over, T. Accelerating advances in continental domain hydrologic modeling. Water Resources Research, 51(12), 10078-10091, 2015.*

*Beck, H. E., van Dijk, A. I. J. M.,de Roo, A., Miralles, D. G., McVicar, T. R.; Schellekens, J., Bruijnzeel, L. A. Global-scale regionalization of hydrologic model parameters. Water Resources Research, 52, 3599-3622, 2065*

*Beck, H. E., van Dijk, A. I. J. M., de Roo, A., Dutra, E., Fink, G., Orth, R., and Schellekens, J.: Global evaluation of runoff from 10 state-of-the-art hydrological models, Hydrology and Earth System Sciences, 21, 2881-2903, 10.5194/hess-21-2881-2017, 2017.*

*Kauffeldt, A., Halldin, S., Rodhe, A., Xu, C.-Y., and Westerberg, I. K.: Disinformative data in large-scale hydrological modelling, Hydrology and Earth System Sciences, 17, 2845-2857, 2013.*

*Xia, Y., Mitchell, K.; Ek, M., Cosgrove, B., Sheffield, J., Luo, L., Alonge, C., Wei, H., Meng, J., Livneh, B., Duan, Q., Lohmann, D. Continental-scale water and energy flux analysis and validation for North American Land Data Assimilation System project phase 2 (NLDAS-2): 2. Validation of model-simulated streamflow. Journal of Geophysical Research, 117, D03110, doi:10.1029/2011JD016051, 2012.*

*Zhang, Y., Zheng, H., Chiew, F. H. S., Arancibia, J. P., and Zhou, X.: Evaluating Regional and Global Hydrological Models against Streamflow and Evapotranspiration Measurements, Journal of Hydrometeorology, 17, 995-1010, 10.1175/jhm-d-15-0107.1, 2016.*

---

## Author Comment (AC2) · 5 Jul 2018

**Text in bold: Reviewer comments**; *Text in italic: Authors response*

**Guy Schumann - Referee #2**

**This paper describes in detail a much needed continental-scale cross-comparison study at the continental scale of the typical regional MGB model using not only different global scale models but also observed or observation inferred variables (e.g. TWS from GRACE, satellite altimetry). The paper is technically very sound and strong and I did not see any problems with the methods employed.**

*We thank Dr. Schumann for dedicating his time to reviewing our manuscript and for highlighting the need of such a study to the scientific community.*

**I really enjoyed reading this paper and although it is fairly long in places, I think it is written in a very comprehensive way and very well organized and presented - I applaud such work and writing. Well done! This said, there are some main points I would like to highlight and see addressed before publication.**

*Thank you very much for this motivating comment. We did our best efforts to draw the attention of a broad public, as well as to extend a regional model to the continental domain using interesting approaches of global-scale modeling.*

**I think it would benefit the paper a lot by listing a number of steps or recommendations to follow for large-scale hydrologic model assessment or validation**

*Thanks for the suggestion. We will add some recommendations. It may be also interesting to highlight South American basins that can serve as "stress tests" for hydrological models, linking with the comments from Reviewer #1.*

**It looks to me as though generally speaking the headwaters are difficult to get right or better said "to agree with other models", which means to me that they are generally very difficult to model correctly. This is of course not surprising given that the topographic complexity and hydrological processes in these regions are not well represented in the models. It would be useful if the authors could comprehensively outline the reasons for those "problem areas".**

*Thanks for the suggestion. We will add some additional discussion about these problem areas. Indeed, there is first an issue of scale. Global or even continental models are not designed to provide estimates for headwater catchments and small rivers due to the resolution of these models and datasets used. Methods for downscaling / interpolation of forcing data can also impact model results, in addition to limitations of satellite estimates.*

**As far as I understand the authors, model calibration is still challenging and therefore could also be responsible for explaining some or even most of the differences observed between different models. Logically it follows that there**

**should be the general recommendation to define and build a set of data that should be used for calibration of large scale models, so that comparison studies later are even more valuable. I think the authors, if they can agree, should call for such a data set in their section of "Model adjustment" (section 3.3) or later in the conclusion is maybe even a better place.**

*Thanks for the suggestion. We can add a recommendation on joining efforts to set up a continental dataset for South America, which can facilitate the intercomparison / validation of models with scales ranging from regional to global.*

---

## Author Response (AR1)

**Toward continental hydrologic-hydrodynamic modeling in South America - Revised Manuscript**

5 **Editor Comment (EC): Comments to the Author:**

**Dear Authors,**

**Two reviewers judged your manuscript. One reviewer was very positive and one reviewer has some doubts about the scientific significance while rating the scientific quality as good.**

10 **However, they both (especially reviewer 1) provide valuable suggestions for improvement and clarifications. Please take into account all these points raised by the reviewers and provide an updated manuscript for further review.**

*Dear Editor and Reviewers (Anonymous Referee #1 and Dr. Guy Schumann)*

15 *We appreciate the feedback on our paper and the constructive comments and suggestions that improved the quality of the revised manuscript. The response to both referee comments can be found in the following sections. We submitted a tracked changes version of the manuscript to highlight parts of the text where modifications were made with respect to the previous version. Authors' comments on the revised manuscript are indicated by numbers that match specific questions of the referees (e.g., RC#1-3*

20 *refers to the comment 3 by reviewer #1). In addition, we ask for permission to include a few additional changes that are marked in the revised manuscript as well.*

Text in bold: Referee comments (RC); *Text in italic - Authors response (AC)*

**Anonymous Referee #1**

5 **RC#1-1: This m/s presents a continental hydrological model for South America. It is forced by global rainfall and climate data and calibrated to streamflow records for a large number of stations. The agreement with recorded streamflow is presented, as well as that with satellite-derived evaporation (ET) and total water storage (TWS). The agreement with observed streamflow is better than that of an ensemble of 3 global models driven by the same precipitation**
10 **estimates.**

**Overall assessment: This appears an overall competent and sound study, but I am missing some truly new scientific insights. The abstract suggests the main insights are (1) calibrating rainfall-runoff parameters is necessary to simulate discharge appropriately; and (2) implementing hydrodynamic routing is also important. I don't think either of those is really very novel. I do not**
15 **think there was ever any doubt that parameter calibration against streamflow records was going to improve the agreement with those same records (noting that the "appropriately" used in the abstract is obviously a subjective term, or at least one that would have to be purpose-specific). The second conclusion also hardly seems surprising and has been shown in previous studies, specifically for the Amazon basin. Indeed, the authors provide several literature references that**
20 **offered those very conclusions.**

**On the positive side, I do think this is an interesting study that has the potential to be a valuable contribution. I thought some of the most interesting contributions from this study were:**

**1) There is a much larger set of streamflow gauging stations in existence in South America**
25 **than is represented in global databases and typically used to calibrate global models;**

**2) The use of a large number of altimetry-derived water level records is interesting;**

**3)** **The authors provide some interesting commentary on the hydrological conditions that likely explain consistently poor performance by global models in some of the basins in South America.**

*AC: We are grateful to the reviewer for reading our manuscript and pointing out relevant questions that needed clarification. To the best of our knowledge, this is the first study that provides a comprehensive cross-scale comparison between regional/continental x global models, supported by spatially distributed, daily discharge data. As stated by the reviewer, there is no doubt that calibration against discharge records leads to improvements in this same variable. Intercomparison studies indicate that models with some degree of parameterization generally perform better on average (e.g., Zhang et al. al., 2016, Beck et al., 2017). On the other hand, results from this paper put light on what extent both calibration and improvement in model physics (routing) are expected to improve estimates of daily river discharges, focusing on different aspects (overall agreement, high and low flows, timing, bias) and regions of South America. To our knowledge, no other study provides such discussion.*

*Although these results do not necessarily show the performance limit that can be achieved by current global forcing data, we understand that these results are of interest to the modeling community that has been seeking for locally relevant hydrological estimates, especially in under development regions. Recent studies call for cooperation between scales (e.g., Archfield et al., 2015) and this work is going exactly on that direction. In summary, continental-scale modeling shows that it is possible to get better discharge estimates by using global data and methods that are currently available, as well as knowledge and methods developed for the studied region. Therefore, we believe that our manuscript brings new information that is relevant in the context of regional, continental and global-scale hydrological modeling.*

*To address the referee comments about scientific insights, we made changes in the abstract, introduction and conclusions (please, see the comments on the revised manuscript indicated by RC#1-1). Some of these changes were made by reinforcing the contributions highlighted by the reviewer in his/her 3 point statements above.*

**RC#1-2: I was somewhat surprised that the majority of forcing and spatial parameterisation approaches used for the "regional" MGB model were, in fact, the same as used for global models. Furthermore, Fig. 11 appears to suggest that the inclusion of hydrodynamic routing was a minor factor in explaining the generally better performance. Therefore, it would seem that the larger number of streamflow gauges and their good use in a more intensive calibration were the real reasons for better performance. That in its own right is useful, as it sets a benchmark that global models should be able to achieve with appropriate parameter calibration.**

**What is unclear, however, is whether that would go at the detriment of the agreement with other observations of the water cycle, such as ET and TWS. It is common that a heavy emphasis on streamflow calibration leads to deterioration in other terms. Therefore, I was surprised that the authors did not include the global models in their comparison against ET and TWS, to assess whether those were simulated better or worse. (The altimetry water levels are less relevant in such a comparison, as one would assume that better discharge simulation also produces better water level simulation. Nonetheless, a comparison with the global models might still have been of interest.)**

*AC: Thanks for the comment. Yes, many of the techniques and databases used to prepare the continental model are similar to those used by global models. Such strategy indicates that the latter can achieve similar results even without a significant increase in the number of computational elements (We added this comment in the conclusions of the revised paper). As pointed out by the reviewer, parameterization has a greater impact on model results in comparison to improved routing, although efforts in calibration may be not effective in regions such as the west side of La Plata basin. Another problem is when a simplified routing structure is applied together with calibration focused in gauge stations with large drainage areas, especially in basins affected by floodplains (e.g., Amazon at Obidos). This can cause problems on flow timing at upstream regions as discussed on the current manuscript. We reinforced in the conclusions that both calibration and hydrodynamic routing cannot be neglected if simulation of daily river discharges is desired for this continent, which is more objective.*

*Regarding the comparison between the continental x global models using variables such as ET and TWS the reviewer makes an interesting suggestion, but we understand that this would not fit into the context of the present study. We have focused on river discharge because this variable is widely used for water resources planning and practical applications (e.g., hydrological forecasting, reservoir operation). It would be difficult to demonstrate if a given model has better accuracy than another with respect to ET since the reference (CDR) has large uncertainties. In addition, to our knowledge, TWS for WRR-2 is not directly available for each model, but rather for an ensemble mean generated with the inclusion of several other global models (i.e., not only those used in this study). Yet, even if there is some interest in comparing water level anomalies, this would not be possible because water level is not an output of WRR-2.*

*We agree that calibration against discharge records could lead to decreased performance in other hydrological variables such as ET and TWS. This is the reason why we performed several evaluations of both variables seeking to document model errors and to understand potential sources of uncertainty. Nonetheless, we added a recommendation in the conclusions of the revised manuscript to also include such observation inferred variables in further cross-scale comparison studies.*

**RC#1-3: In summary, the present m/s mainly seems to assert the common "our model is better than theirs", which is not very insightful as it appears almost entirely due to calibration. There are however some good opportunities to make this a more valuable (and cited!) contribution:**

*Thanks for the comment; we would like to highlight some points about this statement. Model intercomparison studies sometimes make direct comparisons of performance (e.g., Xia et al., 2012, Zhang et al., 2016, Beck et al., 2016, Beck et al., 2017), and we understand that this is important because identifying model shortcomings is essential for future improvements to be made. In this context, there is no best model (first because it depends on the objectives) but rather issues of scale and the best use of available information. Based on our findings, we expect that models with a regional domain will have, on average, better performance than continental models even if the former are forced with global datasets. Of course, it would be of high interest to understand which areas/rivers, flow conditions (high*

*and low flows, bias, timing...) and for what reasons this happens, which definitely represents a reduction of the gap between different scales.*

*Nevertheless, we have made several point changes in the revised manuscript to reasonably suppress the "model A is better than model B", making sure that statements are more constructive (please, see*
5 *comments on the text indicated by RC#1-3). We believe that the text is sounding better now.*

**RC#1-4: Propose these model simulations, along with the station and altimetry records, as a benchmark for global models by making them directly available online to the global modelling community. To make the MGB model acceptable as a benchmark for an allround hydrological**
10 **model, you should demonstrate whether the global models are also less effective in simulating ET and TWS. This would provide insight into whether only the streamflow simulations can be considered benchmark, or the other water cycle components as well.**

*Thanks for the suggestion. Model outputs and supporting files are now available for public access in a*
15 *specific website, which can be found in "data availability" at the end of the revised manuscript. In addition, we have made available a list of the gauge stations in the supplementary material, as these data can be downloaded according to the links in section 3.2.1. We have also recommended the development of a dataset to facilitate both model validation and intercomparison in South America (conclusions), as suggested by Reviewer #2.*
20 *With respect to ET and TWS, we have previously indicated the reasons why it would not be feasible to make such a comparison at this moment, apart from the fact that the current study is already long. Therefore, we changed the focus in the introduction from "benchmark to global models" to other contributions than can be achievable within a cross-scale model intercomparison study.*

25 **RC#1-5: Provide more discussion and emphasis on the understanding of the hydrological conditions of some of the "problem" basins, so that they might become a valuable "stress test" of hydrological model performance.**

*AC: Thanks for the suggestion. We added commentary on this topic in the conclusions of the revised manuscript.*

*AC: Thanks for the suggestion. We have included additional discussion about these problem areas at the end of section 4.2.*

**RC#2-5: As far as I understand the authors, model calibration is still challenging and therefore could also be responsible for explaining some or even most of the differences observed between different models. Logically it follows that there should be the general recommendation to define and build a set of data that should be used for calibration of large scale models, so that comparison studies later are even more valuable. I think the authors, if they can agree, should call for such a data set in their section of "Model adjustment" (section 3.3) or later in the conclusion is maybe even a better place.**

*AC: Thanks for the suggestion. We added a recommendation on joining efforts to set up a continental dataset for South America, which can facilitate the intercomparison / validation of models with scales ranging from regional to global. This modification was made in the conclusions of the revised manuscript.*

**Toward continental hydrologic–hydrodynamic modeling in South America**

[revised manuscript text omitted]

**[V2] Comentário:** RC#1-1: We changed this part in the abstract (details about performance in different regions were kept only in the conclusions) to emphasize the main contributions of our work.

[revised manuscript text omitted]

**[V3] Comentário:** RC#1-4: As the term "benchmark" here directly suggests that A is better than B, we changed the text to emphasize the "filling the gap" idea between modelling scales.

**[V4] Comentário:** RC#1-1: We have made changes here in order to clarify the objectives of our work, focusing on the potential contributions of different modeling scales.

[revised manuscript text omitted]

**[V6] Comentário:** RC#1-3: Changed to avoid non-constructive statements regarding different modeling scales.

**[V7] Comentário:** RC#1-3: Changed to avoid non-constructive statements regarding different modeling scales.

main stem of the Orinoco basin. In terms of NSE, the largest differences in performance previously observed for the KGE now extend to the main Amazon River, to its tributaries in the eastern region (i.e., Tapajos and Xingu) and also to both Magdalena and Tocantins–Araguaia basins, with values of d_NSE ≥ 0.8. With respect to low to medium flows (d_NSElog), there is a similar pattern to d_KGE (except for East Brazil), although with more pronounced differences in the Amazon and Magdalena regions.

The Ensemble GM performs relatively well in all statistics over temperate regions with the absence of lowland rivers (e.g., southern Brazil and Southern Chile),  and outlines specific locations where the continental approach may be somewhat limited. For example, correlation is slightly reduced for the continental model (–0.1 < d_r < –0.2) in areas over the Parnaiba basin and Chile, while a marked decrease in timing performance (d_Abs (DI) < –20 days) is observed in dry Argentinian rivers like Salado (southwest of La Plata basin) and Desaguadero (Colorado basin). Poor estimates of river geometry and large overestimation of flows in these regions may be causing excessive flooded areas and consequent peak attenuation. Regarding intermediate to low flows, considerable differences in model performance are observed mainly over specific rivers in East Brazil and parts of Amazon basin near the Andes Cordillera (d_NSElog < –0.8), as well as in regions over South Chile (–0.2 < d_NSElog < –0.6) that are potentially affected by snowmelt.

(Figure 10)

Table 3 shows differences in median discharge statistics for each global model and also for the Ensemble GM in comparison to the MGB continental model. Because LISFLOOD and WaterGAP3 account for reservoir impacts in their model structure, gauge stations with naturalized discharge data ($n = 98$) were excluded from the analysis to provide a fair assessment.  Except for the Ensemble GM, differences  in performance regarding each pair of models (global × continental) are quite similar for KGE (~0.45) and NSElog (~0.5), while being highly variable for both NSE (~1 to ~1.8) and bias (~4 % to ~30 %). Differences in median DI are between 1 and 2 days, which can be important for cases where flood timing is around this order of magnitude. Among the estimates from global models only, the Ensemble GM outperforms four out of the six metrics analyzed (KGE, NSE, NSElog and DI) with correlation (d_r = 0.03) equivalent to the best of global models for this metric (LISFLOOD, d_r = 0.02). A reduction in performance occurs only when bias is evaluated, where 50 % of the gauge stations have an absolute difference equal to or greater than 11 % compared with differences in HTESSEL/CaMa-Flood (d_Abs (BIAS) ≈ 8 %) and WaterGAP3 (d_Abs (BIAS) ≈ 4 %). In the assessment by Beck et al., (2017a) for basins < 10 000 km² around the world, LISFLOOD also had an advantage in correlation when compared with other global models, while WaterGAP3 demonstrated problems related to baseflow index, which may be indicated here by the largest difference of NSElog (d_NSElog ≈ 0.6).

**[V8] Comentário:** RC#1-3: Changed to avoid non-constructive statements regarding different modeling scales.

**[V9] Comentário:** RC#1-3: Changed to avoid non-constructive statements regarding different modeling scales.

[revised manuscript text omitted]

**[V14] Comentário:** RC#1-1: We included some numbers from the cross-scale comparison that may be relevant (as well for timing below).

**[V15] Comentário:** RC#1-2 and RC#1-3: We have rewritten this statement at the end of the current paragraph in order to improve the contribution between modelling scales.

**[V16] Comentário:** RC#1-2: To emphasize that parameterization and hydrodynamic routing not just improve, but cannot be neglected in South America to simulate daily discharges.

**[V17] Comentário:** RC#1-2: Emphasizing calibration as the main responsible for the large improvement, as indirectly suggested by the reviewer

**[V18] Comentário:** RC#2-3: we added here as a recommendation for large-scale evaluation studies.

**[V19] Comentário:** We removed this to put emphasis on the most relevant contributions of the study.

continental dataset that can be exploited by a broader audience, thus contributing to reduce the gap between regional and global modeling communities.

~~, Among the largest basins in South America, challenges still remain in simulating the La Plata (in particular, the Paraguay River) and this is a major concern given its economic importance for many countries of the continent. In this case, improvements in representing discharges are expected after conducting a more detailed river discretization in the Pantanal region and inclusion of a quasi 2D connection scheme over its floodplain (see Fleischmann et al., 2018). Using our online approach (i.e., fully coupled hydrologic–hydrodynamic) together with routing water in multiple downstream directions will enable both representing diffuse flows over the floodplain and feedback between surface water and soil processes, which can be very pronounced over large, seasonally flooded tropical wetlands such as the Pantanal (Paz et al., 2011, 2014).~~

Finally, the results found in this study  show that extending a regional, fully coupled hydrological-hydrodynamic model to the entirety of South America is feasible. This underscores the importance of  regional knowledge, which can  indicate relevant hydrological processes and datasets to be included in continental/global model simulations. We hope that moving from regional toward continental hydrologic–hydrodynamic modeling will bring new opportunities for operational practices such as real-time hydrological forecasting, which is the topic of an ongoing research. Nevertheless, several improvements should be carried out in the model structure not only to achieve a better understanding of the underlying processes but also to provide further insights about human impacts on South American water resources. This includes the representation of reservoirs, lakes and water abstractions. Uncertainties in model parameters are also important to be addressed and should be further investigated.

*Data availability*. Results from the MGB model are available  to the public at http://www.ufrgs.br/lsh. All other datasets used in the present study can be accessed using the websites cited in this manuscript.

*Author Contributions*. V. A. S. worked on the MGB model for South America, designed the study, performed model calibration/simulations, managed all datasets (in situ, remote sensing and global models) and produced the manuscript; R. C. D. P. helped on study design, MGB code optimization, interpretation of results, contributed to writing of the manuscript and conducted paper review; A. S. F. and F. M. F. assisted on MGB calibration and conducted paper review; A. L. R. helped with the interpretation of results. P. R. M. P. provided support with GIS routines and MGB simulations over La Plata basin; A. P. and S. C. provided support with satellite altimetry data and associated analyses; W. C. was the research supervisor, contributed to writing of the manuscript and conducted paper review.

*Competing interests*. The authors declare that they have no conflict of interests.

**[V20] Comentário:** RC#1-5, RC#2-3 and RC#2-5: We added commentary here as requested by both reviewers (problem areas/stress tests, recommendations for large-scale model evaluation; call for datasets).

**[V21] Comentário:** We removed this part since relevant information about La Plata basin is already discussed in the paragraph above

**[V22] Comentário:** RC#1-3: Changed to avoid non-constructive statements regarding different modeling scales.

**[V23] Comentário:** RC#1-1: Changed to emphasize the "filling the gap" between modelling scales.

**[V24] Comentário:** RC#1-4: Added here a link for accessing MGB outputs.

[revised manuscript text omitted]

[V25] Comentário: We updated this figure to provide RMSE instead of NSE metric, to be in agreement to the text at section 4.1.4.

**Figure 9: Comparison between MGB and GRACE (JPL RL05 v2 mascon solution) monthly TWS anomalies for major South American basins.**

[Figure]

**Figure 10: Difference between performances of MGB and the Ensemble GM for discharge metrics. Values considered as not significant (gray) are within the ranges –0.05 to +0.05 (d_r, d_KGE, d_NSE and d_NSElog), –5 % to 5 % (d_Abs(BIAS)) and –2 to +2 days (d_Abs(DI)).**

[Figure]

Figure 11: Boxplots of global (above center line) and continental (below center line) model performances for different South American regions. MGB model configurations: hydrodynamic routing with calibrated rainfall-runoff parameters (MGB_HD_calib), hydrodynamic routing with uncalibrated rainfall-runoff parameters (MGB_HD_noCalib), non-hydrodynamic routing with calibrated rainfall-runoff parameters (MGB_noHD_calib) and non-hydrodynamic routing with uncalibrated rainfall-runoff parameters (MGB_noHD_noCalib). Gauge stations with naturalized flows were removed from the analysis to provide a fair comparison.